# Regression-based season-ahead drought prediction for southern Peru conditioned on large-scale climate variables

Eric Mortensen[1], Shu Wu[2], Michael Notaro[2], Stephen Vavrus[2], Rob Montgomery[3], José De Piérola[4], Carlos Sánchez[4], Paul Block[1]

[1]Department of Civil and Environmental Engineering, University of Wisconsin–Madison, Madison, 53706, USA
[2]Nelson Institute Center for Climatic Research, University of Wisconsin–Madison, Madison, 53706, USA
[3]Montgomery Associates Resource Solutions LLC, Cottage Grove, 53527, USA
[4]Southern Peru Copper Corporation, Santiago de Surco, Lima 33, Peru

*Correspondence to*: Paul Block (paul.block@wisc.edu)

**Abstract.**

Located at a complex topographic, climatic, and hydrologic crossroads, southern Peru is a semi-arid region that exhibits high spatiotemporal variability in precipitation. The economic viability of the region hinges on this water, yet southern Peru is prone to water scarcity caused by seasonal meteorological drought. Meteorological droughts in this region here are often triggered during El Niño episodes; however, other large-scale climate mechanisms also play a noteworthy role in controlling the region's hydrologic cycle. An extensive season-ahead precipitation prediction model is developed to help bolster existing capacity of stakeholders to plan for and mitigate deleterious impacts of drought. In addition to existing climate indices, large-scale climatic variables, such as sea surface temperature, are investigated to identify potential drought predictors. A principal component regression framework is applied to eleven potential predictors to produce an ensemble forecast of regional January-March precipitation totals. Model hindcasts of 51 years, compared to climatology and another model conditioned solely on an El Niño-Southern Oscillation index, achieve notable skill and perform better for several metrics, including ranked probability skill score and a hit-miss statistic. The information provided by the developed model and ancillary modelling efforts, such as extending the lead time of and spatially disaggregating precipitation predictions to the local level as well as forecasting the number of wet/dry days per rainy season, may further assist regional stakeholders and policymakers in preparing for drought.

## 1 Introduction.

Southern Peru is a semi-arid region just north of the Atacama Desert, located at a complex topographic, climatic, and hydrologic crossroads. With elevations ranging from sea level to over 6,000 meters, the area is a patchwork of snow-capped Andean mountains, highlands and plateaus, and large expanses of coastal desert. Due to its proximity to the Amazon rainforest, the Atacama Desert, and the Pacific Ocean, the climate patterns that govern the region's precipitation vary considerably, both seasonally and annually. Although a notable portion of this region drains to Lake Titicaca, which is itself a part of a larger endorheic basin, the majority of the region's water flows into the Pacific Ocean through networks of small rivers and quebradas, or seasonal creeks. While the topographic, climatic, and hydrologic factors of the region produce spatiotemporal variability in

the distribution of water resources (Tapley and Waylen, 1990), southern Peru can generally be characterized as water scarce (Alegría, 2006; Kuroiwa, 2007; Ugarte, 2012; Chinchay Alza, 2015).

Nonetheless, southern Peru displays a high economic dependence on activities driven directly by water availability, specifically agriculture and mining (Higa Eda and Chen, 2010). The region is home to some of the nation's richest olive and grapevine fields, as well as several large-scale copper mining operations. Both of these industries are heavily dependent on water consumption. Additionally, several large urban areas such as Arequipa, Juliaca, and Tacna (Fig. 1) necessarily require large quantities of water to thrive as economic and cultural centers.

Although not equivalent (Van Loon, 2015), meteorological drought in this region often directly translates into hydrologic drought. Droughts, like the one that struck in early 2016, have a critical impact on the success and survival of the region. During that year, agricultural outputs of southern Peru were reduced by up to 75% (ANA, 2016), necessitating the creation of an emergency contingency fund for impacted farmers by Peru's national water authority (in Spanish, Autoridad Nacional del Agua, or ANA). ANA also declared states of emergency for two cities, Tacna and Arequipa. Consequentially, the cities' water supplies were reduced by more than one-fourth. The mining operations of the region were also negatively impacted, with ANA ordering mining companies such as Southern Peru Copper Corporation (SPCC), to reduce their water consumption and, transitively, copper production, resulting in lost economic potential and reduced fiscal resources for the region as a whole.

The severity of this most recent bout of drought, unfortunately, is not unprecedented; other droughts in the past have also caused serious economic and social consequences. The drought event of early 1983 was particularly severe across southern Peru (Caviedes, 1985). Before this event, hazard preparedness essentially did not exist in Peru. The drought, which coincided with deadly flooding in the northern part of the country, was met with slow and uncoordinated official disaster relief. Even after Peru developed their national hazard preparedness program following this event, the region continued to be vulnerable to drought. In 1998, an estimated $200 million in direct losses occurred over the southern Andes of Peru due to drought (Lavado-Casimiro et al., 2013).

The challenges created by drought, when combined with other factors such as cultural differences and socioeconomic disparities, can instigate economic instability and societal stress regionally (Lynch, 2012). While tools exist to monitor drought and hydroclimatic conditions in Peru, such as the Peruvian Drought Observatory (ANA, 2014), drought prediction remains a relatively unexplored field for southern Peru. If droughts could be predicted several months or seasons in advance, regional decision makers, private entities, local interests, and other stakeholders may be able to reduce their immediate vulnerability to hydroclimatic variability (Sadoff and Muller, 2009). Season-ahead drought prediction may afford stakeholders more capacity to address mid- and long-term water resources planning goals (Ugarte, 2012; Chinchay Alza, 2015).

We develop a statistically-based season-ahead principal component regression (PCR) model that predicts seasonal precipitation totals to address this existing gap. The PCR model draws on a far-reaching pool of large-scale climate variables that influence southern Peru precipitation through ocean-atmosphere teleconnections. The model is evaluated against climatology and simpler Niño index-based models to understand if the inclusion of several predictors leads to more skillful prediction, particularly for dry years in this drought-sensitive region. In addition, several ancillary applications of this model are explored, including lead time extension, spatial disaggregation, and wet/dry day frequency prediction, in an attempt to provide further detailed information which may be relevant to stakeholders.

## 2 Data Description.

Monthly precipitation data are available for 29 stations distributed across the region over a period of 51 years (1966-2016; Fig. 1). Six of the 29 stations are owned and operated by SPCC, with the remaining stations belonging to Peru's national meteorological service (in Spanish, Servicio Nacional de Meteorologia e Hidrologia del Peru, or SENAMHI).

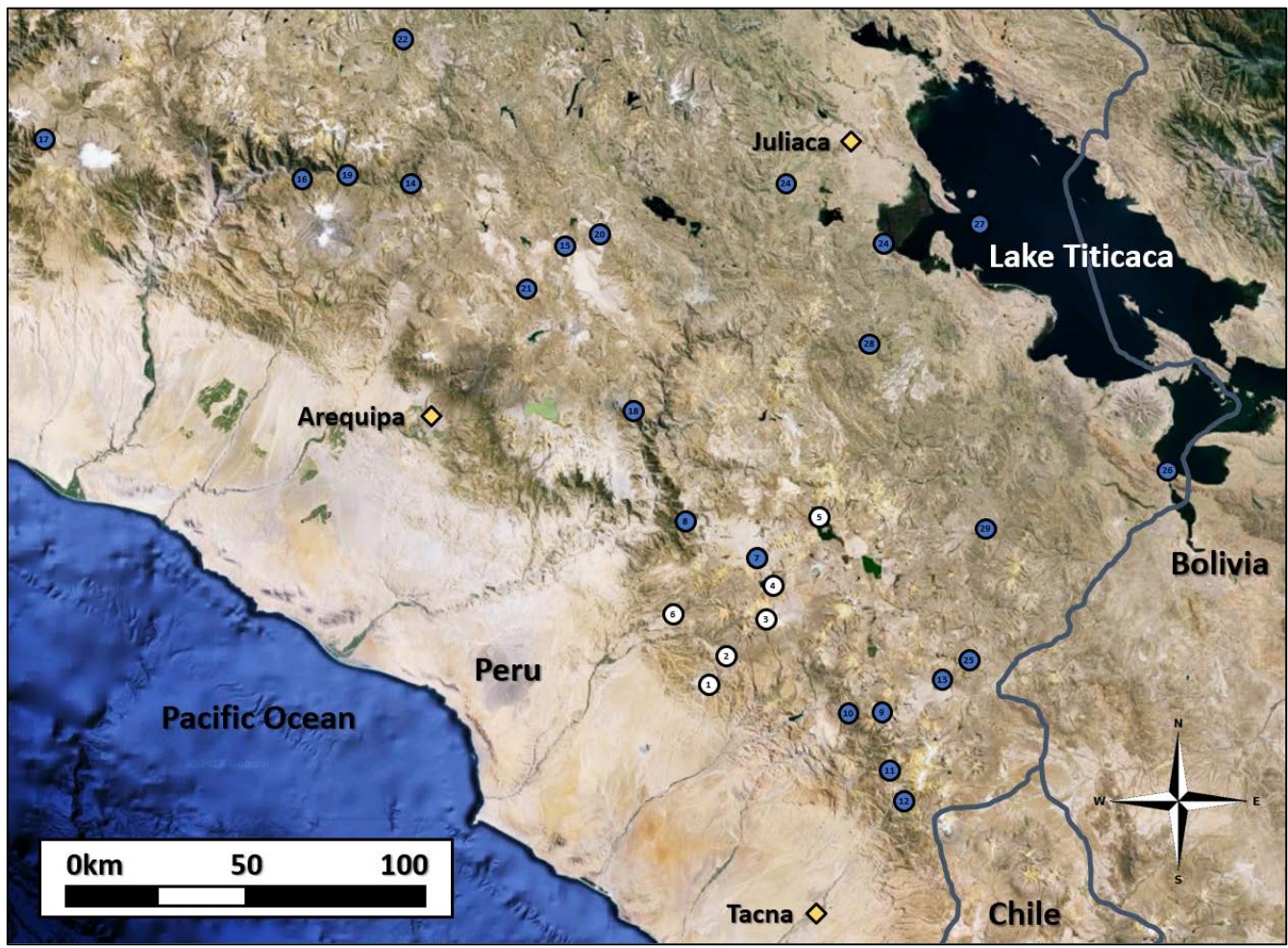

**Figure 1: White circles represent locations of SPCC stations; blue circles represent SENAMHI stations. Three major urban centers are labeled and stations are numbered from 1-29 (map generated using Google Earth imagery and station information from SPCC).**

The 29 stations provide spatial coverage for an area of 65,000 km² and are located in a variety of environments, including the edge of the Atacama Desert, the islands of Lake Titicaca, the dry grassy plains of the Altiplano, and the mountainous terrain of the Central Andes. The topography of the region is noteworthy. While the 29 stations considered in the study cover an elevation range from 3,100 m to 4,600 m (Fig. 2, mean elevation 3870 m), this portion of southern Peru ranges from sea level at the Pacific Ocean to over 6,000 m in the high Andes.

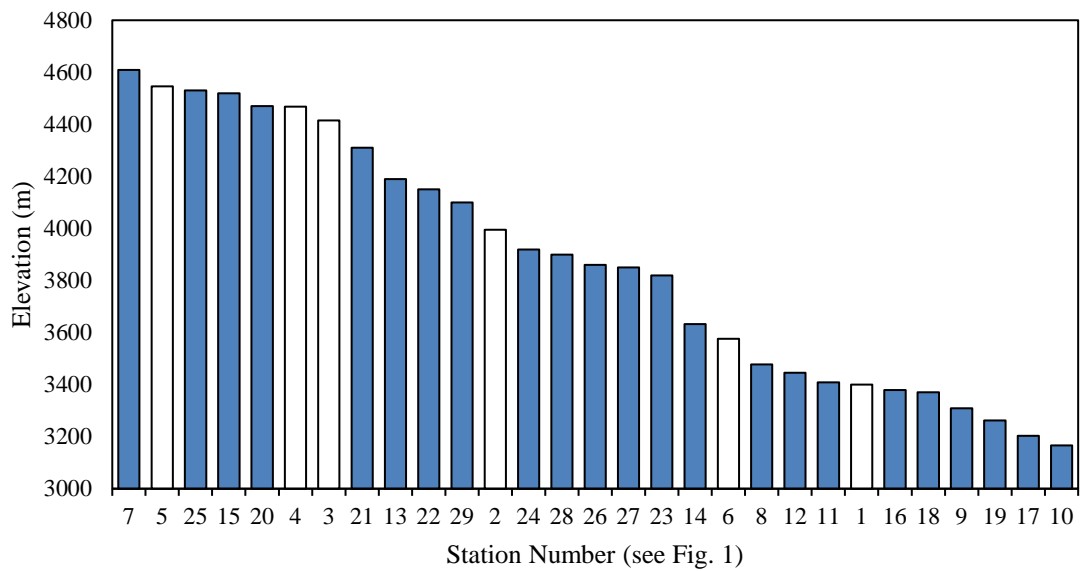

**Figure 2: Elevations of all 29 stations included in the study. Bars are numbered and colored in accordance with Fig. 1, with white bars representing SPCC stations and blue bars representing SENAMHI stations.**

Cross-correlations between all of the stations were calculated based on available monthly precipitation totals (average Pearson's correlation coefficient, r=0.92). For any missing station data (<1% of total data), the ten most highly correlated stations were identified, and multiple regression based on monthly statistics was used to interpolate a probable missing value. In most cases, high correlation coefficients between estimated missing points and observed data suggest that this simple methodology is effective in filling the data.

Potential large-scale climate predictors, including sea surface temperature (SST), sea level pressure (SLP), and geopotential height (GH), were retrieved from the National Oceanic and Atmospheric Administration (NOAA) Earth System Research Laboratory Physical Sciences Division (ESRL-PSD). The data are based on National Centers for Environmental Prediction-National Center for Atmospheric Research (NCEP-NCAR) reanalysis data, version 1 (Kalnay et al., 1996) and are available as monthly average on a 2.5° x 2.5° global grid (this dataset is available in full from 1948 to present). The specific regions and periods of the aforementioned climate variables considered in this study are listed in Table 1 of Section 4. In addition, ESRL-PSD monthly/seasonal climate correlation and composite mapping tools are used in this analysis.

In addition to the aforementioned large-scale climate variables, several established teleconnection indices, such as Niño 3.4 (Rayner et al., 2003), Pacific Decadal Oscillation (PDO; Mantua et al., 1997), North Pacific index (NP; Trenberth and Hurrell, 1994), and Western Hemisphere Warm Pool (WHWP; Wang and Enfield, 2001), are evaluated in this study.

### 3 Southern Peru Rainy Season and Large-scale Climate Influences.

In the mid-high elevation regions of southern Peru, as in most tropical zones, the annual cycle is dominated by a wet and dry season (Fig. 3). For southern Peru, the rainy season occurs from November to April (Kuroiwa, 2007); however, the majority of precipitation in the region occurs during January, February, and March (JFM; 315 mm on average). JFM precipitation represents, on average, more than two-thirds of annual precipitation for the region, with some locations receiving up to 85% of annual precipitation during the three-month period. This precipitation is crucial to the region's economic activities and environmental stability. During the rainy season, for example, surface reservoirs and underground aquifers are replenished for multi-sectoral water resource use during the dry conditions that characterize the rest of the year. These rains also directly impact the phenology of many wild plants and agricultural operations, and are intrinsically tied to the function of seasonal creeks that drain the region. As mentioned, severe and wide-reaching economic, environmental, and societal consequences can be realized in an abnormally dry rainy season. Thus, JFM is identified as the season of interest for this study.

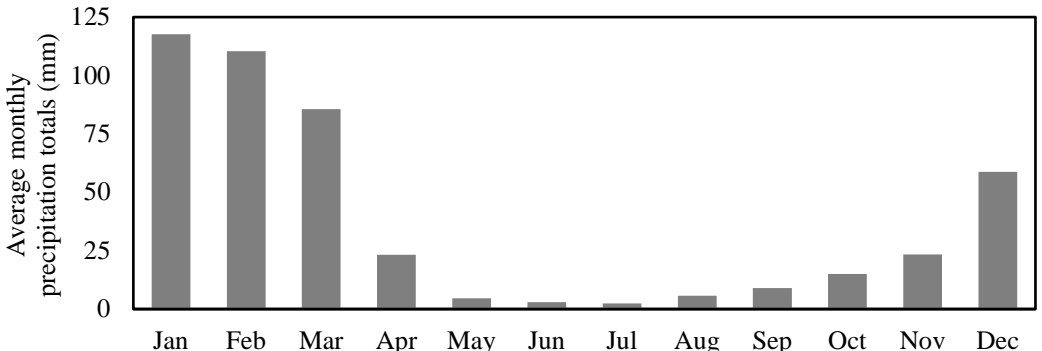

**Figure 3: Average monthly precipitation (mm) of 29 stations southern Peru.**

To evaluate the spatial and temporal patterns of regional precipitation, a principal component analysis (PCA) is performed on JFM seasonal precipitation totals (von Storch and Zwiers, 2001) based on data from the 29 stations. In PCA, a dataset is decomposed into orthogonal, uncorrelated modes representing distinctive signals, or variance, present in the dataset. PCA yields information describing both spatial patterns (empirical orthogonal functions, EOFs) and temporal trends (principal components, PCs) of variance experienced in the dataset.

Even with significant changes in elevation across the region, the sign of the first EOF spatial pattern of all stations is negative (and at similar magnitudes) generally implying spatial homogeneity (Eklundh and Pilesjö, 1990; Ogallo, 1980; Mallants and Feyen, 1990; Bisetegne et al., 1986) of JFM seasonal precipitation within this relatively small region. Additionally, the first PC of the precipitation time series captures 51% of the temporal variance in the data, and correlates well with station-averaged JFM seasonal precipitation observations (r = 0.99; Fig. 4).

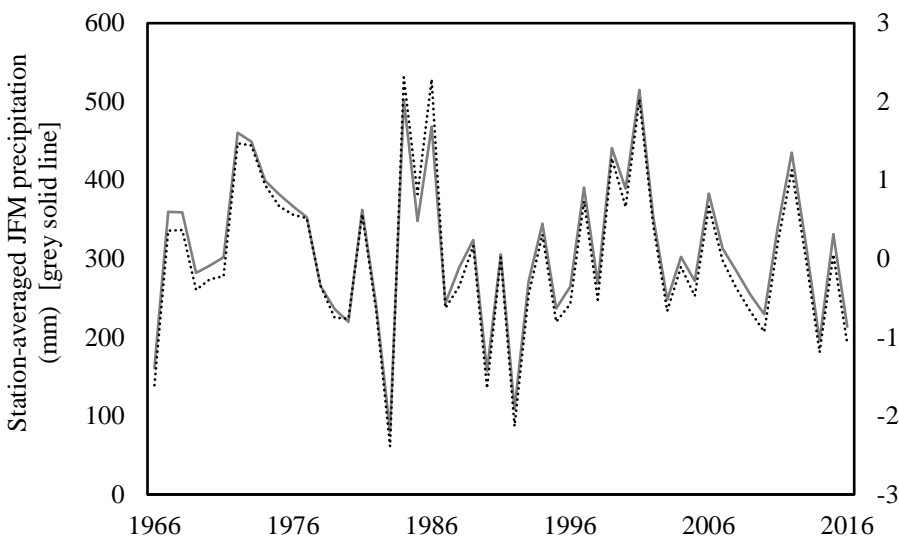

**Figure 4: Station-averaged JFM precipitation (mm) and the first PC anomalies for the period of record, 1966-2016, using data from 29 precipitation stations (introduced in Sect. 2).**

This exceptional level of correlation between the averaged observations and the first PC (as well as high levels of correlation between this first PC time series and individual station data) suggest that the station-averaged time series is an appropriate representation of regional precipitation. It should be noted that the second mode captures an additional ~20% of variance, with the third dropping to ~5%. These three PCs represent a cumulative 75% of variance experienced in the dataset (it is assumed

that the remaining PCs describe only minor or spurious variance). To identify physical mechanisms that modulate precipitation that result in the observed temporal variance, the complex regional climate system must be comprehensively analysed.

During the rainy season, the tropical Southern Hemisphere receives increased amounts of solar radiation that destabilizes the atmospheric boundary layer, inducing deep convection and moist air advection (Vuille et al., 1999; Garreaud, 1999). This

directly translates to increased levels of evapotranspiration in the Amazon basin, with moisture transported deep into the atmosphere by a complex network of deep convection systems, including the upper level of the Bolivian High (Lenters and Cook, 1997). In general, the winds associated with this deep convection are easterly and northerly, carrying moisture towards the Andes from the Amazon (Fuenzalida and Rutllant, 1987; Chaffaut et al., 1998; Rao et al., 1996; Vizy and Cook, 2007). The Andes induce an orographic effect in which more precipitation occurs at windward locations and higher elevations of the

region (Garreaud, 1999). Meanwhile, the precipitation at the leeward (western) side of the mountain range and lower elevations is markedly reduced; this region of southern Peru exists in the rain shadow of the Andes, a fact especially relevant for the study. Instead of an abrupt switch between wet and dry conditions as might be expected by some other notable rain shadows

in the world, the Altiplano (and the majority of the stations used in this study) exists in a transitionary zone of sorts and exhibits a gradual wet to dry gradient from northeast to southwest.

Previous studies have identified SST anomalies in the equatorial Pacific as a substantial factor impacting regional precipitation patterns in southern Peru (Vuille et al., 2000; Garreaud et al. 2003; Espinoza Villar et al. 2009; Lavado-Casimiro et al., 2013; Cid-Serrano et al., 2015). This area of the Pacific is commonly associated with the El Niño-Southern Oscillation (ENSO) phenomenon, and several studies further identify the SST domain of 5° N-5° S, 120° W-170° W, known as Niño 3.4 (Trenberth, 1997), as particularly influential in modulating JFM precipitation. Strong El Niño (warm SST) conditions in the Niño 3.4 region are typically associated with drought in southern Peru, whereas La Niña (cool SST) conditions often align with wetter-than-average conditions (Fig. 5, El Niño and La Niña thresholds of 0.5°C and -0.5°C, respectively, included for context).

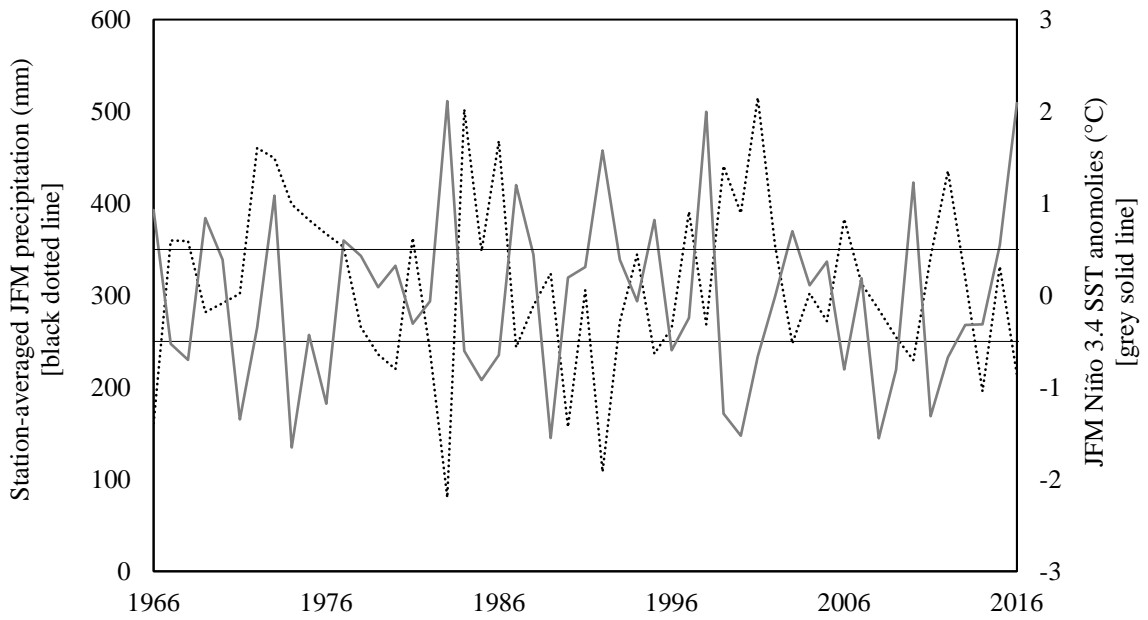

**Figure 5: Station-averaged JFM precipitation and concurrent JFM Niño 3.4 SST anomalies (r=-0.57, p-value = 0.000013). El Niño and La Niña thresholds marked with black solid line. During the period of record, 13 JFMs exceeded the El Niño threshold (0.5°C) and 18 JFMs the La Niña threshold (-0.5°C).**

Furthermore, prior studies have determined that droughts in southern Peru are not generally caused by limited moisture availability, but rather limited moisture transport. During El Niño episodes, enhanced upper-level westerly flow from the Pacific Ocean weakens the typical wind patterns of the region, blocking easterly winds laden with moisture that normally falls as precipitation in southern Peru (Garreaud et al., 2003; Takahashi, 2006). During La Niña, easterly flow is enhanced, often resulting in greater precipitation and cloud cover, and lower temperatures in the central Andes (Vuille, 1999).

The phase and strength of ENSO does not necessarily correspond to a specific outcome for seasonal precipitation, a fact particularly evident in three notable cases (bolded and underlined, Fig. 6). In late 1972, a strong El Niño developed off the coast of South America; however, instead of expected dry conditions, JFM 1973 surprisingly turned out to be one of the wettest rainy seasons on record for the region (Garreaud et al., 2003). In contrast, ENSO index values indicative of neutral to weak La Niña conditions prior to JFM 1990 and 2014 would have typically been interpreted to mean normal to slightly wetter-than-average conditions, yet these years are two of the driest rainy seasons on record.

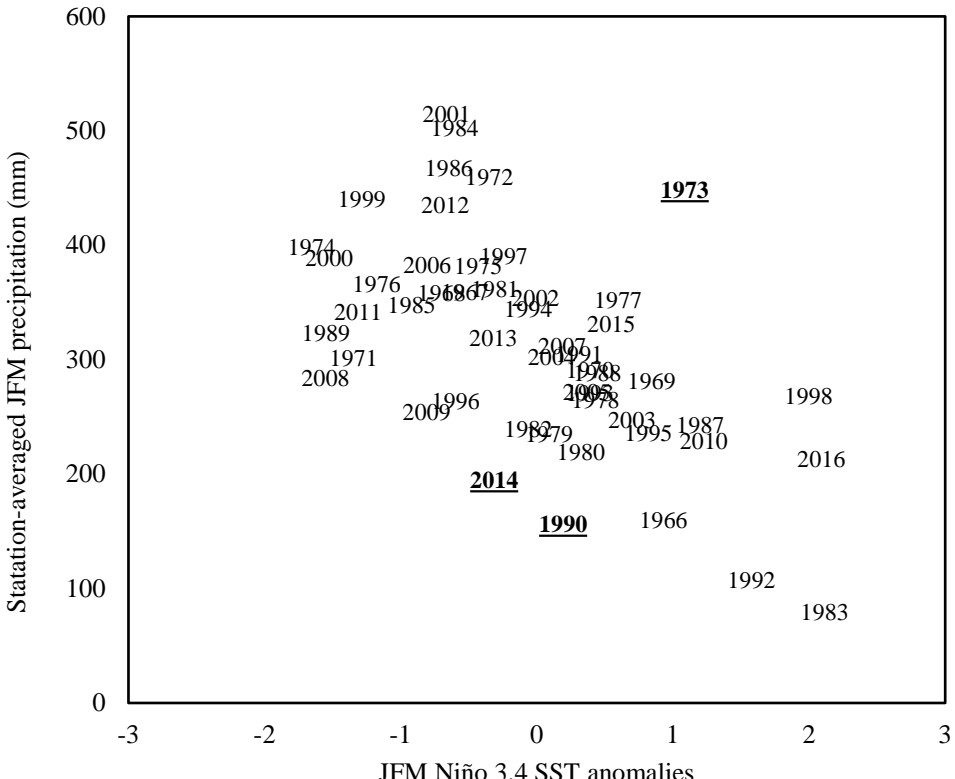

**Figure 6: Three outlier years in which general relationship between Niño 3.4 and regional precipitation did not hold to be true are bolded and underlined. If the three outlier years (1973, 1990, and 2014) are removed, the relationship between precipitation and SST anomalies strengthens (r=-0.66, p-value = 0.000009).**

Such deviations from the generally understood relationship between ENSO and regional JFM seasonal precipitation are likely due to other climate phenomena, and support the two-fold notion of ENSO's influence on seasonal precipitation as well as the presence of additional climatic factors that modulate the region's precipitation. Regions and variables of interest highlighted

in other studies (and considered in this study) as mechanisms potentially modulating precipitation in southern Peru include Tropical Atlantic SST, several SST regions of the Pacific, and the Bolivian High.

While the main moisture source for Altiplano precipitation is the tropical lowlands to the east of the Andes, this moisture ultimately originates over the trade wind regions of the Tropical Atlantic (Vuille et al., 2000), the primary source of moisture to the Amazon. In particular, SST anomalies in the North Tropical Atlantic regulate dry season precipitation anomalies in the western Amazon (Marengo et al., 2008; Zeng et al., 2008; Yoon and Zeng, 2010; Fernandes et al., 2011). When the North Tropical Atlantic is anomalously warm, the Intertropical Convergence Zone shifts northward, causing net water vapor divergence, anomalous subsidence, and reduced precipitation in western/southern Amazon (Marengo, 1992; Marengo et al., 2008; Yoon and Zeng, 2010), and southern Andes (Lavado-Casimiro et al., 2012).

In the Pacific Ocean, locations outside of the traditional ENSO region also appear to impact precipitation in this region of South America. Although the subtropical Pacific is immediately adjacent to the region of interest, it typically contributes little moisture to southern Peru because low-level zonal flow and associated moisture from the sea is blocked by steep regional terrain and large-scale subsidence (Rutllant and Ulriksen, 1979). The Pacific Ocean, however, still plays a significant role in controlling the regional hydrologic cycle due to these zonal winds. The Pacific Decadal Oscillation (PDO) has also been identified as modulating precipitation throughout much of South America (Enfield, 1996; Kayano and Andreoli, 2007). This multi-decadal, low frequency oscillation of North Pacific SST impacts several regional climate systems and has been widely accepted by the hydrometeorological community as being distinct from ENSO (Deser and Blackmon, 1995; Mantua and Hare, 2002; Wang et al., 2008). Additionally, the Western Hemisphere Warm Pool (WHWP), a region of abnormally warm SST off the coast of Central America with lobes in the Caribbean and Pacific Ocean, may likewise influence regional precipitation as a result to the warming cycle's impact on rainy season precipitation in equatorial Central and South America via trade wind modulation (Wang and Enfield, 2003; Wang and Enfield, 2006). Finally, the North Pacific (NP) index, which describes SST and SLP variability in the North Pacific, has a direct connection with changes to Tropical Pacific SST and circulation patterns (Trenberth and Hurrell, 1994). PDO, WHWP, and NP indices are all considered in this study.

The upper-level Bolivian High, located over the Altiplano during December-April, is related to latent heat release over the Amazon (Silva Dias et al., 1983; Lenters and Cook, 1997). The position and strength of the High has been linked to precipitation anomalies over the Altiplano during the rainy season. Specifically, a weakened, northward shifted Bolivian High is often associated with persistent dryness on the Altiplano (Aceituno and Montecinos, 1993; Lenters and Cook, 1999; Vuille et al., 2000), whereas a strong, southward shifted Bolivian High favors deep convection on the Altiplano and increased moisture availability (Garreaud and Aceituno, 2001; Garreaud et al., 2003). Thus, the position of the Bolivian High impacts zonal winds during the Altiplano's rainy season; dry (wet) conditions over the Altiplano are associated with anomalous westerly (easterly) flow in the region (Aceituno and Montecinos, 1993; Lenters and Cook, 1999).

While the first EOF of regional precipitation likely illustrates ENSO's influence on regional precipitation, it is possible that higher order modes may describe other climatic and topographic forcings such as interconnected large-scale climatic phenomena or observed orographic effects. For example, the second EOF exhibits a dipole pattern, which may be related to the rain shadow phenomenon that causes the northeastern portion of the region to be wet and southwestern to be dry.

## 4 Identification of Seasonal Precipitation Predictors.

Potential predictors of JFM precipitation are identified by analyzing persistent large-scale and local climate variables in the prior season of October–December (OND) based on the suite of variables and indices identified in Section 3, and validated through correlation mapping, composite mapping, and global wavelet analysis. The purpose of these three methods is to identify climate variables and indices that partially explain the variance in JFM precipitation and as a result may serve as potentially skillful predictors in the development of a season-ahead precipitation prediction model.

Spatial correlation maps between the first three PCs of JFM regional precipitation (explaining approximately 75% of the variance) and global OND climatic variables, including SST, SLP, and GH at 200 hPa, illustrate regions of correlation and potentially relevant teleconnections. Only December values are used for SLP and GH given their limited atmospheric persistence. For example, the correlation between OND SST and the first PC of JFM regional precipitation produces a pattern emblematic of the classic ENSO phenomenon (Fig. 7). The area near (but not exactly) Niño 3.4 has the strongest correlation (r=-0.54), indicative of a relationship in which, generally, abnormally warm (cool) water in this region corresponds with dry (wet) conditions in southern Peru, supporting previous findings.

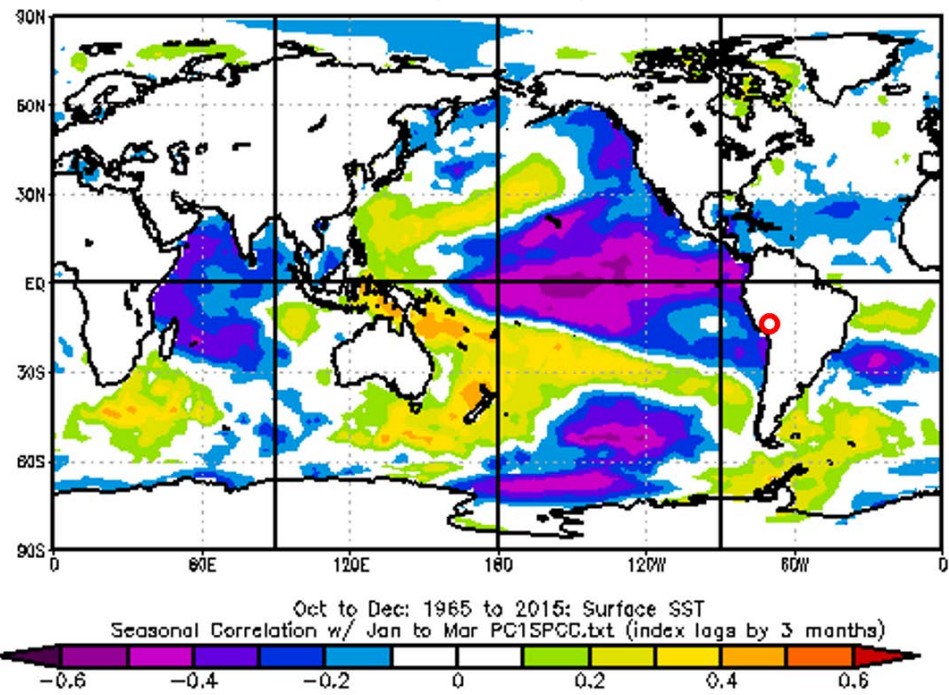

**Figure 7: Correlation between global OND SST and first PC of regional JFM precipitation. Study region identified with red circle.**

Not all regions, however, that display relatively high correlations with the PCs are necessarily physically relevant. To limit spurious correlations, only regions of statistical significance at the 95% confidence level and justifiable (via relevant, peer-reviewed literature) physical influence on moisture transportation to southern Peru are selected as potential predictors. Additional areas of interest identified through spatial correlation mapping include an area of SLP off the western coast of Mexico/USA (roughly 35° N-20° N, 150° W-135° W) and an area of geopotential height above southern Bolivia/northern Argentina (roughly 10° S-15° S, 70° W-65° W). These two areas, in addition to the mentioned region of SST in the equatorial Pacific, display statistical significance at the 95% confidence level to at least one of the three analyzed PCs. We speculate that these two regions of high correlation likely have a physical relation to the WHWP and Bolivian High, respectively.

Composite maps illustrate climate conditions for a single period or subset of periods, and may be especially useful for understanding forcing mechanisms in anomalous periods. For example, OND SST for the nine subsequent driest JFM seasons on record for southern Peru subtracted from OND SST for the nine subsequent wettest JFM seasons on record for southern Peru produce large positive anomalies in the equatorial Pacific Ocean. This composite map (Fig. 8) further indicates the potential importance of ENSO in explaining JFM precipitation variability in the study region.

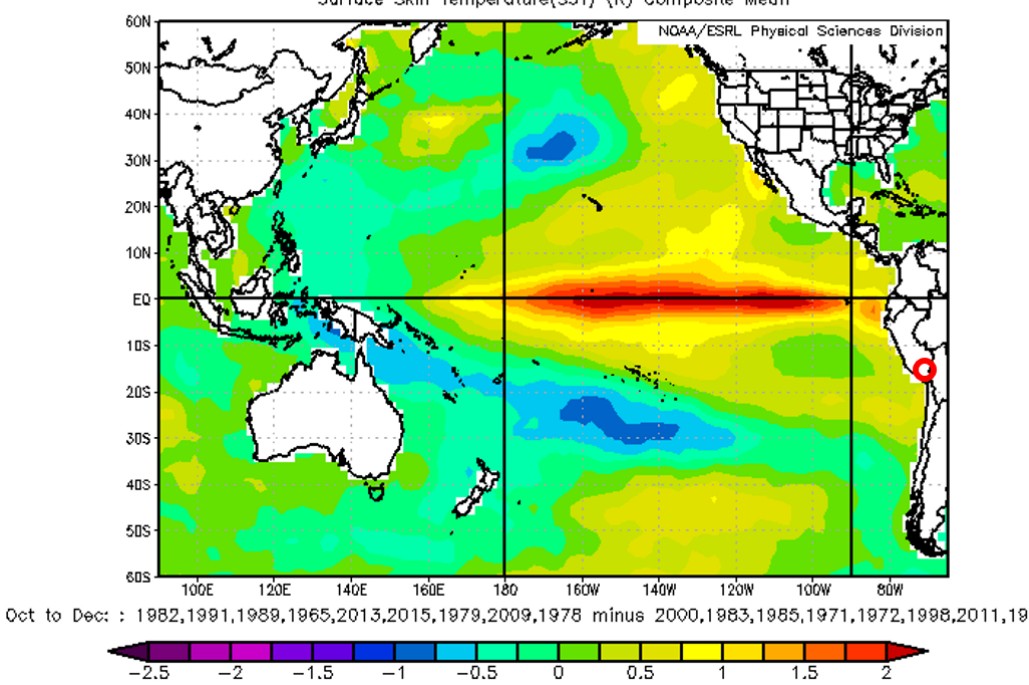

**Figure 8: SST conditions for OND seasons following the nine driest JFM seasons (1983, 1992, 1990, 1966, 2014, 2016, 1980, 2010, and 1979) subtracted from SST conditions for OND seasons following the nine wettest JFM seasons (2001, 1984, 1986, 1972, 1973, 1999, 2012, 1974, and 1997). Study region identified with red circle.**

Additional composite maps, namely subsets of years with the strongest El Niño and La Niña years or years with wetter-than-average El Niño years and drier-than-average La Niña years, led to identification of ENSO, SST gradients in the North Pacific and Tropical Atlantic Oceans, and the Pacific lobe of the WHWP as potentially skillful predictors of JFM precipitation. Interestingly, for deviations from the typical ENSO-precipitation relationship (i.e., dry- vs. wet-El Niño JFMs and dry- vs.

10  wet-La Niña JFMs), the resulting anomalies in the North Pacific as well as WHWP appear to be similar in size and magnitude. Thus, during the unexpectedly wet 1973 JFM or unexpectedly dry 1990 JFM, for example, these two SST regions may have modulated the effect of other large-scale climate variables, such as equatorial Pacific SST, on regional precipitation.

Finally, wavelet analysis is applied to the observed station-averaged JFM precipitation totals to identify different frequency

15  signals that may exist in the dataset. More specifically, wavelet analysis is mainly used to detect the changing of dominant periods with time. Wavelet analysis decomposes a time series into time-frequency space to identify significant modes of variability and illustrate how variability may change with time (Torrence and Compo, 1998). Using a Morlet 6.00 transform (Morlet et al., 1982) on the station-averaged JFM precipitation time series, signals at a ~3-5-year band, ~12-16-year band, and ~24-year band are identified as statistically significant at the 95% confidence level (Fig. 9).

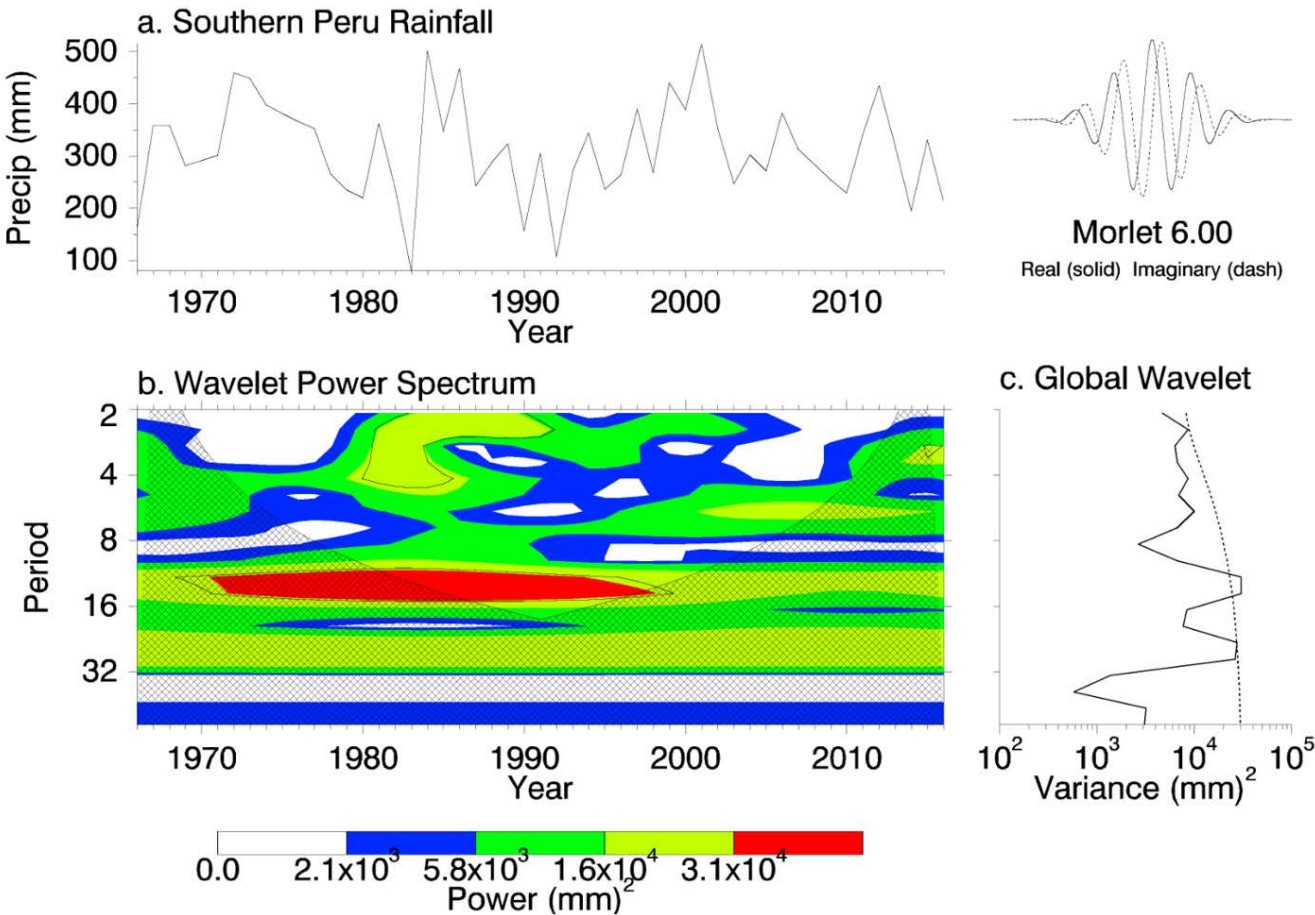

**Figure 9: (a) Precipitation time series, (b) statistically significant signals at T = ~3-5, ~12-16, and ~24 years (statistically significant periods at 95% confidence level outlined), (c) global wavelet variance with 95% confidence level delineated by dotted line.**

The identified signals at ~3-5 years and ~12-16 years are likely indicative of ENSO and perhaps PDO, respectively. These identified underlying periodicities of the precipitation data further affirm the inclusion of large-scale climate indices with both relatively short and long periods of oscillation. Occasionally, wavelet spectrum analysis can artificially amplify the power of longer periods. To determine whether the ~24-year signal is truly statistically significant, further testing, such as a Fourier
10 power spectrum, may be warranted (Wu and Liu, 2005), but not undertaken here.

In total, 11 potential predictors are identified for prediction of station-averaged JFM precipitation based on previous literature and inference from spatial correlation maps, composite maps, and global wavelet analysis (Table 1). These potential predictors include both established climate indices and relevant regions of SST, SLP, and GH (as well as gradients of these variables).

All potential predictors included in the model framework display a statistically significant correlation with at least one of the first three PCs of the station-averaged precipitation time series. In addition, five potential predictors are also statistically significant correlated with the station-averaged times series of precipitation, and marked with asterisks in Table 1.

5 **Table 1: The suite of potential predictors for JFM precipitation; correlations are based on JFM total precipitation and spatial averages across the regions noted, with statistically significant correlations marked with an asterisk.**

| Name | Large-scale climate variable | Time frame | Spatial region | | Corr. w/ JFM precip. | Most Correlated PC (r) |
|---|---|---|---|---|---|---|
| Niño 3.4 | SST | OND | 5° N-5° S | 170° W-120° W | -0.53* | PC1 (-0.52) |
| PDO | SST | OND | all areas north of 20° N | | -0.19 | PC2 (-0.35) |
| NP | SLP | D | 65° N-35° N | 160° E-140° W | -0.18 | PC3 (0.28) |
| WHWP | SST | OND | 28° N-8° N | 110° W-40° W | -0.16 | PC3 (-0.32) |
| | SST | OND | 0° -5° S | 160° W-140° W | -0.54* | PC1 (-0.54) |
| | SLP | D | 35° N-20° N | 150° W-135° W | 0.15 | PC2 (-0.36) |
| | SST gradient | OND | 0° -15° S (25° S-40° S) | 15° W-35° W (15° W-35° W) | 0.30* | PC2 (-0.29) PC3 (0.28) |
| | SST gradient | OND | 50° N-40° N (35° N-30° N) | 150° W-135° W (180° -165° W) | 0.38* | PC3 (-0.37) PC2 (0.27) |
| | GH 200 hPa | D | 10° S-15° S | 70° W-65° W | -0.35* | PC1 (-0.31) |

## 5 Methods.

Statistical forecasts have been developed and evaluated for many applications globally, although more effort is still focused on the application of dynamical model predictions; however, there are numerous advantages for selecting statistical models
10 for season-ahead precipitation prediction over other methods involving global atmospheric general circulation models (GCMs), most notably reviewed by Xu (1999). These include GCMs' inability to represent sub-grid features and dynamics and mismatches between GCMs' strengths and hydrology needs (atmospheric vs surface), both of which statistical models can address. Dynamical models are exceptional tools for macroscale climate modeling, but can struggle at the local scale, like that of our project area. Additionally, the complex topography of the region complicates the use of GCMs for regional predictions.
15 For this case, the merits of statistical models appear to outweigh those of dynamical modes.

## 5.1 Principal Component Regression-based Prediction Model.

A PCA coupled with a multiple-linear regression model construct, otherwise known as PCR, is used to predict station-averaged JFM seasonal precipitation for the study region. In this case, the method used to develop the model is advantageous because it accounts for the multi-collinearity present among several of the identified potential predictors (von Storch and Zweirs, 2001) and standardizes all resulting PC values so as not to cause any unintentional preference. After a PCA is performed on the set of area mean values of the identified potential predictors, the PCs are fit to a multiple-linear regression, given as:

$$y = \beta_0 + \beta_1 x_1 + \cdots + \beta_n x_n + e \tag{1}$$

where $y$ is the observed JFM total precipitation, $\beta_0$ is a constant, $\beta_1...\beta_n$ are coefficients, $x_1...x_n$ are the PCs, and $e$ is the error term. Coefficients are determined using the ordinary least squares method (Helsel and Hirsch, 2002).

To create a parsimonious model and minimize overfitting, the optimal number of PCs (i.e. predictors) is selected using the generalized cross-validation (GCV) skill score (Walpole et al., 2012; Block and Rajagopalan, 2007), given as:

$$GCV = \frac{\sum_{t=1}^{N} \frac{e_t^2}{N}}{(1-\frac{m}{N})^2} \tag{2}$$

where $N$ is the number of data points (JFM seasons in the study), $e_t$ is the prediction error or residual (the difference between model predictions and observations), and $m$ is the number of PCs retained as predictors. GCV scores are computed for each model iteration (models with varying numbers of PCs retained), with the preferred model having the lowest GCV score. Models that overfit may have smaller prediction errors, but are penalized for having a larger number of predictors.

After selecting the optimal number of PCs to incorporate into the model, a drop-one cross validation prediction framework is applied to the 51 years of available data. The cross-validated predictions are assembled into a hindcast for the entire period of interest. This includes – for each year of the hindcast – dropping the predictor data (Z) from the year being hindcasted, forming new PCs (and EOFs) conditioned on the remaining years, and fit to observations using multiple regression, providing an intercept coefficient, regression coefficients, and error term (Stone, 1978). The predictor data (Z) from the year dropped are then projected onto the EOFs to provide PCs for the dropped year. Finally, these PCs are multiplied by the appropriate regression coefficients and added to the intercept coefficient to provide a deterministic precipitation prediction for the dropped year. This is repeated for each year.

To create ensemble hindcasts, error terms from all years are assembled and a distribution is fit (using a kernel density estimator; the distribution is approximately Gaussian). For each hindcast year, 1,000 random draws from the distribution are added to the deterministic precipitation prediction to form an ensemble.

### 5.2 Extended Lead Time of Predictions.

In the initial version of the model, predictors are drawn from OND, such that predictions may be issued on January 1st for JFM precipitation. This information may be too late for certain stakeholders (e.g. farmers) who have already made critical decisions regarding their operations and short- to mid-term output goals. Extending the prediction lead time is explored by evaluating progressively earlier 3-month periods. For example, shifting the predictor season to SON, a JFM precipitation prediction wouls instead be issued on December 1st. The potential predictors for each lead time analyzed are identified in
similar fashion to that of the OND predictor season model.

### 5.3 Spatial Disaggregation of Predictions.

Although seasonal predictions of station-averaged regional precipitation may benefit planning at a larger scale, such as by regional water councils or federal entities, more localized predictions of precipitation may prove to be advantageous for sectoral decision-making (mining, farming, etc.). To address this, spatial disaggregation of predictions from the regional-level
to the station-level is evaluated. Using the regional-level categorical prediction probabilities for each year (above normal, near normal, and below normal; Fig. 10), ensemble predictions for each station are generated based on that station's own climatology. For example, the categorical probabilities at the regional-level for 2016 are predicted as 2% above normal, 7% near normal, and 91% below normal. For each station, JFM precipitation observations from all other years (excluding 2016) are randomly selected 1,000 times from that station's JFM precipitation distribution conditioned on the regional probabilities.
Thus, the ensemble of predictions for that station for 2016 will have approximately 91% of its members from the below normal category, 7% near normal, and 2% above normal.

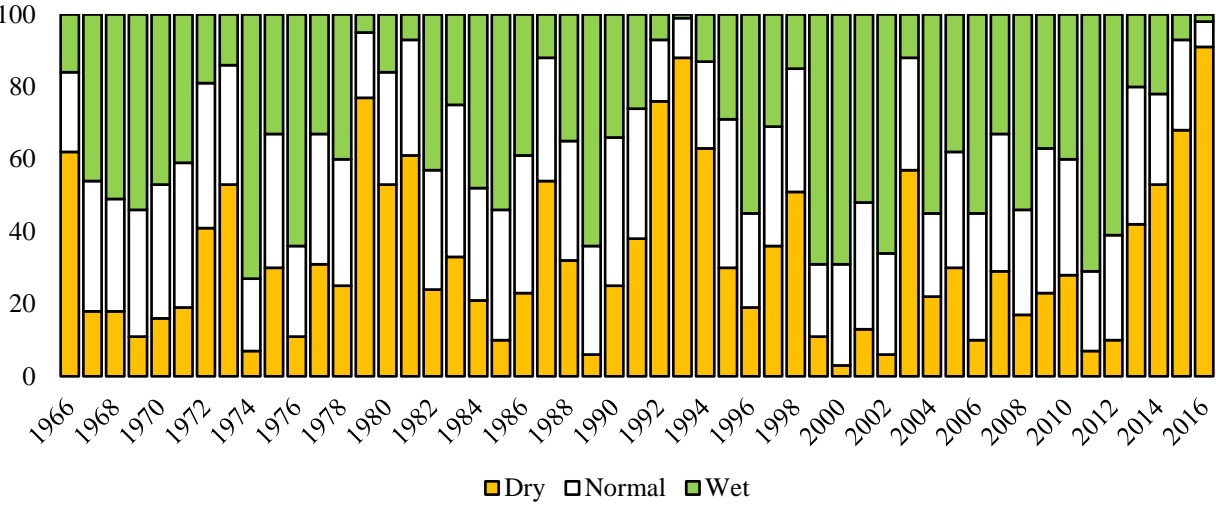

**Figure 10: Categorical probabilities for JFM precipitation totals (y-axis) in each year, as predicted by the regional-scale model.**

The purpose of spatially disaggregating in this fashion is to maintain the statistical integrity of the regional-level prediction while reflecting appropriate magnitudes of precipitation experienced at each station. This methodology ensures that regional- and station-level categorical prediction probabilities match, however absolute precipitation magnitudes across stations may vary significantly (Maraun, 2013). Because the regional and station predictions are related, a dependent t-test for paired samples is applied to test for significant changes or differences between these two. A dependent t-test result of no statistically significant change is desired.

## 5.4 Wet/dry Day Frequency Analysis.

Although predictions of JFM seasonal precipitation totals may be useful for a variety of stakeholders throughout the region, some may prefer additional detailed information such as the frequency of precipitation events expected across a given rainy season. The number of wet or dry days and the intensity of precipitation events can have widespread and serious agronomic/phenologic (Robertson et al., 2008) and infiltration/runoff (Mandal and Nandi, 2017) implications. Such information may also be informative to condition stochastic weather simulators for a wide range of hydrologic or agricultural models (Robertson et al., 2006). To evaluate seasonal statistics of wet/dry day frequency for southern Peru, six of the 29 stations having readily accessible daily data are analyzed. Analogous to the seasonal total precipitation prediction modeling approach, spatial correlation mapping, composite mapping, and global wavelet analysis are all utilized to identify potential predictors describing the expected number of wets days across the JFM season.

**5.5 Model Evaluation.**

The cross-validated ensemble hindcasts and ancillary applications of the model are evaluated deterministically and categorically in this study using three metrics: Pearson's correlation coefficient between observed values and the median of the ensemble forecast; rank probability skill score (RPSS); and a hit-miss statistic presented as contingency tables.

RPSS is based on the ranked probability score (RPS), which measures the categorical accuracy of forecasts (Wilks, 2011). For this study, categories are based on three equal terciles from the observed record (e.g. splitting the ordered observed record into three categories with 17 years in each), and represent above normal (greater than 350 mm), near normal, and below normal (less than 270 mm) total seasonal precipitation. RPS is the cumulative squared difference between categorical probabilities for

forecasted and observed conditions, and takes the form:

$$RPS = \frac{1}{K-1}\sum_{m=1}^{K}[(\sum_{k=1}^{m}f_k) - (\sum_{k=1}^{m}o_k)]^2 \tag{3}$$

where $K$ is the number of categories, $f_k$ is the predicted probability for the $k$th category, and $o_k$ is the observed probability for

the $k$th category (1 if the observation falls in that category and 0 if not). RPS ranges from 0-1, with a perfect forecast scoring 0. RPSS provides the relative improvement of a prediction as compared to a reference prediction – typically climatology (distribution of long-term historical observations), and is given as:

$$RPSS = 1 - \frac{RPS_{forecast}}{RPS_{climatology}} \tag{4}$$

An RPSS value less than zero indicates no forecast skill over the reference climatology forecast (i.e. the forecast model does not outperform climatology). Values greater than zero represent a skillful forecast. A value of one represents a perfect categorical forecast.

The hit-miss statistic describes the occurrence of median model predictions falling into the observed category (above normal, near normal, or below normal conditions). Results are presented in a three-by-three matrix, or contingency table, that illustrates the performance of the model for each category. Contingency tables are an alternative method of assessing the precision of model predictions that relies on categorical probabilities as opposed to simpler methods such as correlation (Svensson, 2016). Of particular interest in this study is the hit rate statistic, or the percentage of time the model accurately predicts (categorically)

the actual observed condition, as well as the double miss rate statistic, or the percentage of time the model makes a two-category error. Because prediction of regional meteorological drought is of particular interest, the likelihood of extremely dry conditions is also considered. For this case, extremely dry conditions are defined as station-averaged JFM precipitation less than 250 mm, which occurs approximately 25% of the time, or during 13 years across the time series.

## 6 Results.

### 6.1 Principal Component Regression-based Prediction Model.

The best performing model, as determined by GCV, includes the first four PCs explaining 83% of the variance in the original potential predictors. The median of the cross-validated, ensemble predictions of JFM precipitation (Fig. 11) correlates with observations at r=0.58.

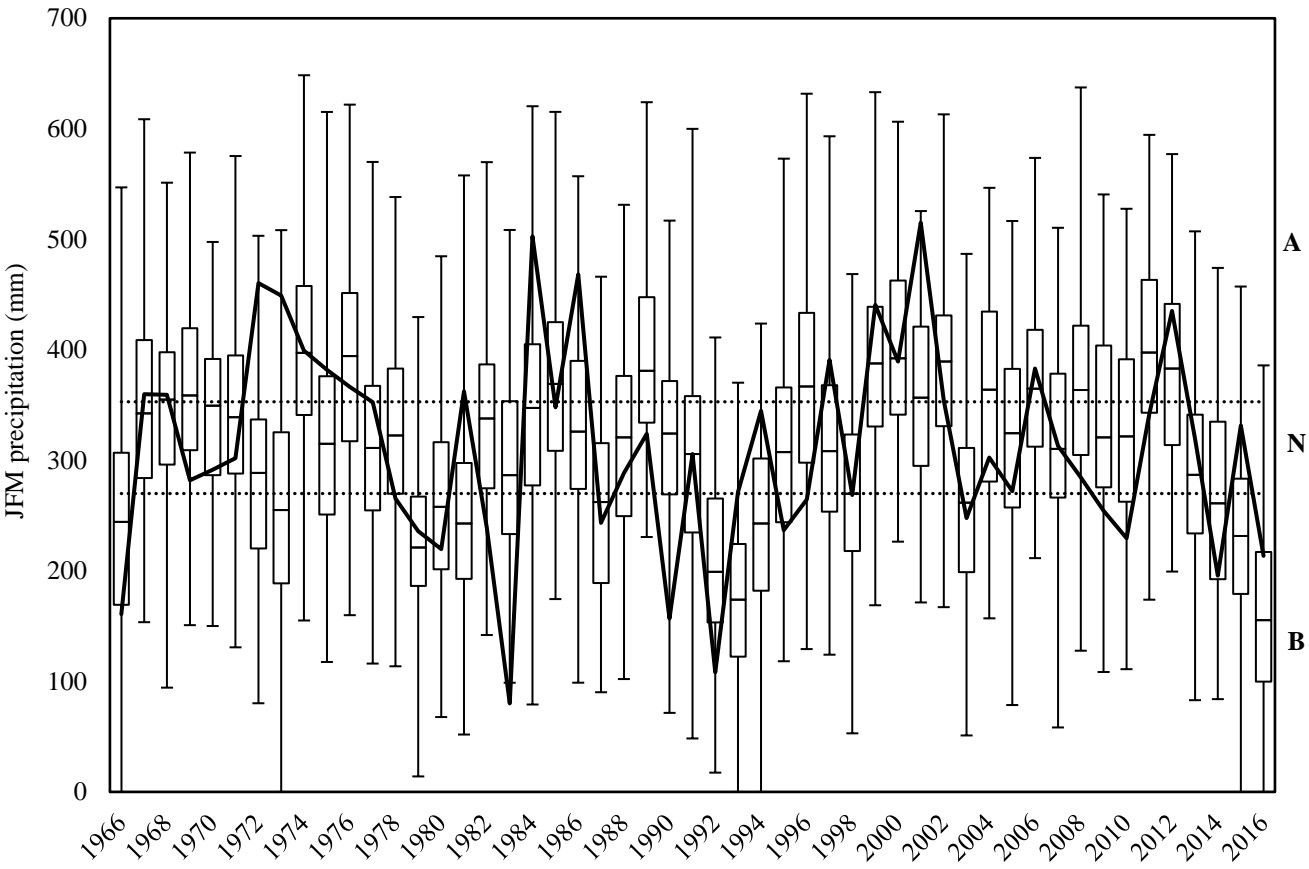

**Figure 11: Box plots of cross-validated, ensemble forecasts of JFM precipitation with observed conditions (solid black line) and categorical thresholds (dotted lines, with delineated categories labeled A, N, and B) included.**

The median RPSS score for the model is 0.16, indicating marginal, yet noteworthy, improvement over climatology. The model also scores a hit rate of 51%, predicting the correct category in 26 of 51 years (Table 2). With specific regard to below normal conditions, the PCR prediction has a 59% hit rate, with 10 of 17 instances correctly predicted.

**Table 2: Hit-miss matrix with three equal categories: above normal (A), near normal (N), and below normal (B) precipitation.**

|  |  | Predicted conditions | | |
|---|---|---|---|---|
|  |  | A | N | B |
| Observed conditions | A | **5** | 9 | 3 |
|  | N | 2 | **11** | 4 |
|  | B | 0 | 7 | **10** |

Above normal (A), near normal (N), below normal (B)

For the 49% of years in which the model missed the observed category, only three times did the model miss by two categories. In all three cases, below normal conditions are predicted yet above normal precipitation is observed (similar to what occurred in 1973). Overall, though, the model has a strong tendency to predict near normal conditions too often (53% of the time versus an expected 33%). It is apparent that the weakest categorical performance is in predicting above normal conditions, with a hit rate of only 29%.

Since drought prediction is of particular interest in this study, an alternative hit-miss metric that uses only two categories – extreme below normal conditions (eB) and above normal/near normal conditions (A/N) – is also evaluated. Here, extreme dry conditions are defined as the lowest quartile of JFMs on record (specifically, 13 years with less than 250 mm of JFM precipitation). The alternative hit-miss metric has a hit rate of 80% in general and accurately predicts 62% of eB conditions (Table 3), a notable improvement compared with the tercile-based hit-miss metric.

**Table 3: Hit-miss matrix with only two categories: above normal/near normal (A/N), and extreme below normal (eB) precipitation.**

|  |  | Predicted conditions | |
|---|---|---|---|
|  |  | A/N | eB |
| Observed conditions | A/N | **33** | 5 |
|  | eB | 5 | **8** |

Above normal/near normal (A/N), extreme below normal (eB)

Overall, model predictions demonstrate moderate skill improvement over predictions conditioned solely on climatology as well as one conditioned on an ENSO index. While a simple linear regression model using OND Niño 3.4 as a sole predictor for JFM precipitation correlates at r=0.53 (only 0.05 less than the more complex PCR model), the RPSS of this Niño 3.4 model is -0.38, or inferior to climatology. Comparing hit-miss metrics, both models perform similarly for tercile-based categories; however, the Niño 3.4 model does not exhibit as much improvement for the two-category assessment (predicting only 23% of

eB years correctly). Both models fail to accurately predict 1973 (unexpectedly wet) and 1990 (unexpectedly dry); however, the PCR model does accurately predict JFM 2014 as dry, even though neutral/weak La Niña conditions existed prior.

## 6.2 Extended Lead Time of Predictions.

For longer leads, no additional predictors were identified as statistically significant for inclusion into the model. In fact, the correlation between JFM precipitation and predictors typically weakened slightly with increased lead time; however, predictions created using potential predictors from extended lead times of SON and ASO are still skillful. Correlations between predicted and observed JFM precipitation only drop slightly (Fig. 12). RPSS also remains positive through ASO.

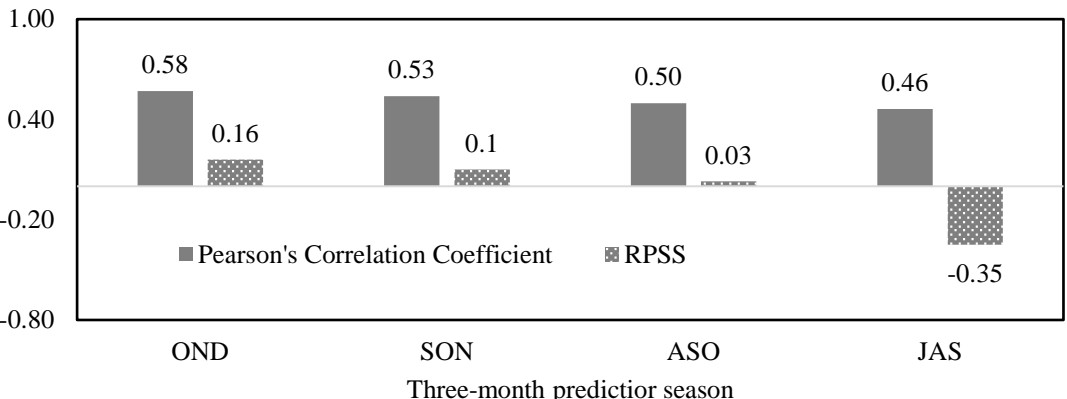

**Figure 12: Correlation coefficients between observed and modeled JFM precipitation and ensemble RPSS for various lead times.**

Upon using the three-month predictor season of JAS, however, RPSS drops below 0, indicating that predictions produced using JAS climate information (and issued on October 1), would have less skill than those produced using simple climatology.

## 6.3 Spatial Disaggregation of Predictions.

None of the stations experience statistically significant decreases in correlation as a result of spatial disaggregation. For two of the stations, correlation values between station-level predictions and station-level observations increase (statistically significant at the 95% confidence interval) as compared with correlation values between regional-level predictions and station-level observations (it should be noted that there is a 5% probability that this happens by chance). The remaining stations exhibit no statistically significant change in correlation as a result of station-level scaling. Five of the stations have significantly improved RPSS values while only one of the stations has a new RPSS value lower than zero. As expected, given identical categorical prediction probabilities, station-level hit scores are nearly identical to regional-level scores (51% overall accuracy), with more accuracy in predicting near normal and below normal conditions.

## 6.4 Wet/dry Day Frequency Analysis.

Interestingly, the first PC of the data captured approximately 85% of the variance in the number of wet days for all six stations with daily data. High correlations are observed between this first PC and SST within the region typically associated with ENSO. No additional regions or climate variables (e.g. sea level pressure, geopotential height, etc.) are identified using this method. Further, composite mapping and global wavelet analysis yield no additional potential predictors for incorporation into the prediction model. Thus, the wet/day frequency model uses only the OND Niño 3.4 season-ahead index as a direct predictor of wet days in any given JFM season. Using the same cross-validation method already described, the number of wet days per season is predicted for each station.

In addition to correlation coefficients between the predicted and observed number of wet days, the average prediction error for above-average and below-average years is reported for each station. Station-specific statistics are listed in Table 4.

Table 4: Correlation values and average absolute errors for predictions of wet days at each station.

| Station (average number of wet days) | Correlation value (r) between prediction and observation | Average absolute error in years with above average number of wet days | Average absolute error in years with below average number of wet days |
|---|---|---|---|
| 1 (25 days) | -0.50 | 10 days | 9 days |
| 2 (36 days) | -0.53 | 11 days | 9 days |
| 3 (51 days) | -0.48 | 11 days | 10 days |
| 4 (54 days) | -0.39 | 12 days | 12 days |
| 5 (54 days) | -0.53 | 8 days | 9 days |
| 6 (33 days) | -0.59 | 10 days | 9 days |

Correlations between predictions and observations range from r=-0.39 to r=-0.59, with the model performing slightly better in predicting the number of wet days in drier years. In general, however, the simple linear model displays an average absolute error ranging between 8 and 12 days.

To consider overall model performance, a hit miss metric quantifies skill in the two previously introduced categories of years (Table 5) – years with above average number of wet days (W) and years with below average number of wet days (D).

**Table 5: Hit miss metric for model predictions of years with above and below average numbers of wet days.**

|  |  | Predicted conditions | |
|---|---|---|---|
|  |  | W | D |
| Observed conditions | W | **105** | 41 |
|  | D | 43 | **103** |

Above average number of wet days (W) and below average number of wet days (D)

Overall, the model correctly predicts whether JFM will have an above or below average number of wet days with an accuracy of ~72%. The model, however, has a notable bias towards over-predicting near normal conditions (Fig. 13).

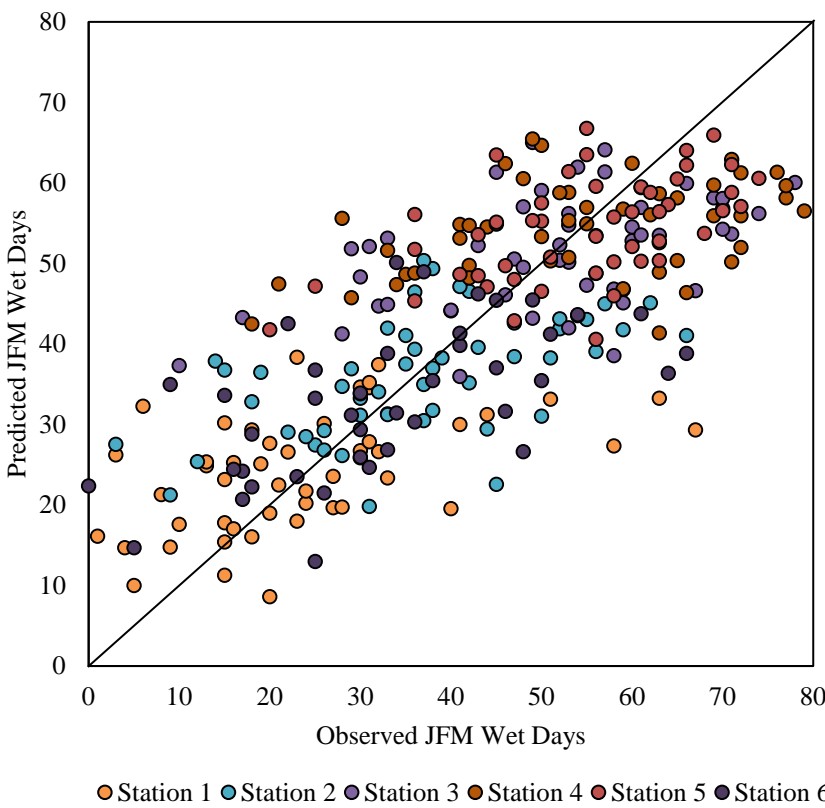

○ Station 1 ○ Station 2 ○ Station 3 ○ Station 4 ○ Station 5 ○ Station 6

**Figure 13: Observed versus predicted number of JFM wet days for six stations across 1966-2016.**

**7 Discussion.**

This study quantifies the value of including additional climate information beyond ENSO-based indices into a prediction model for JFM precipitation in southern Peru. To understand the importance equatorial Pacific SSTs, a second hindcast model is developed using only 9 of the 11 original potential predictors, with Niño 3.4 and SST from 160° W-140° W 0° -5° S dropped. Using the same cross-validated PCR methodology (as well as GCV to determine the optimal number of potential predictor PCs to incorporate, i.e., 3), we produce hindcasts for the period of record (Fig. 14).

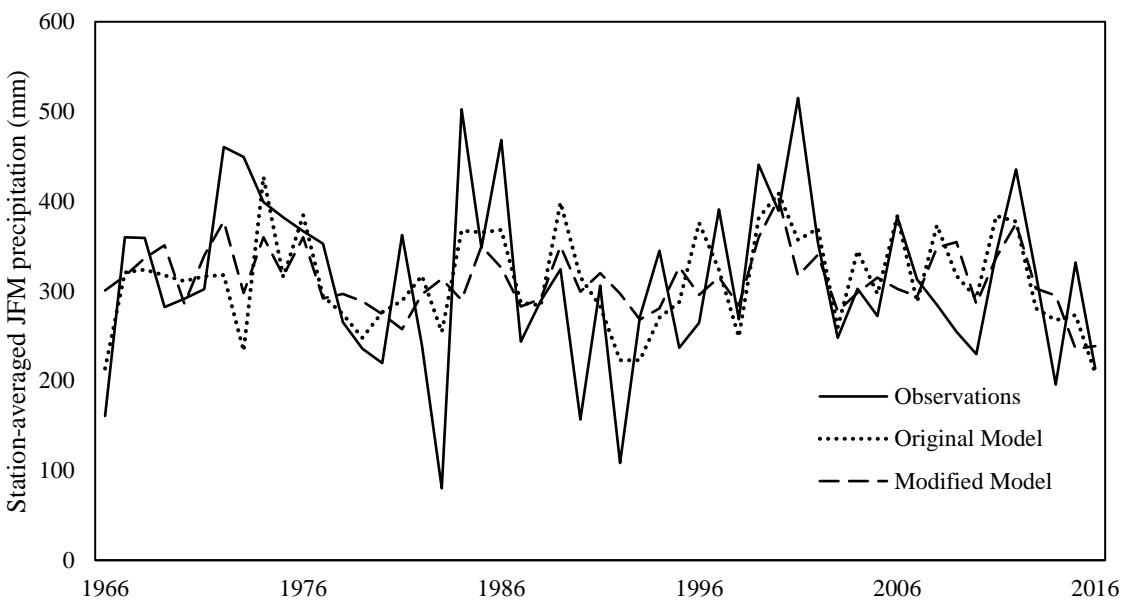

**Figure 14: Observed conditions for the period of record, as well as hindcasts produced using the original model (11 potential predictors, 4 PCs) and the modified model (9 potential predictors, 3 PCs).**

When comparing the results of the modified model to the original model, the importance of including ENSO in a model construct for precipitation prediction in southern Peru is highlighted. For example, the correlation coefficient between predicted conditions and observations drops from r=0.58 to r=0.37 for the original and modified models, respectively (skill reduced by roughly one-third). In addition, RPSS drops to only 0.05% from the original 16%, indicating that the information provided by the modified model is just barely more skillful than climatology. The hit-miss metric also illustrates the diminished skill of the modified model (Table 6).

**Table 6: Hit-miss matrix for the modified model with three equal categories: above normal (A), near normal (N), and below normal (B) precipitation.**

|  |  | Predicted conditions | | |
| --- | --- | --- | --- | --- |
|  |  | A | N | B |
| Observed conditions | A | **6** | 10 | 1 |
|  | N | 0 | **15** | 2 |
|  | B | 1 | 14 | **1** |

Above normal (A), near normal (N), below normal (B)

The modified model displays an evident bias towards predicting near normal conditions (more than 75% of the time). While the hit score of this model is reduced to 43%, more striking is the fact that the modified model produces an instance in which above normal conditions are prognosticated, but below normal conditions are experienced – perhaps a more devastating outcome for this region than a situation with below normal predictions but above normal observations). These metrics reflect the critical importance of including ENSO in regional precipitation prediction.

Model skill remains relatively constant with increasing predictor lead time, notably predictions produced using ASO predictor information for November 1. This additional lead may prove beneficial to stakeholders in the region. For example, in the 2016 drought, ANA made emergency declarations for the cities of Tacna and Arequipa at the beginning of January based on projected water availability. This allowed minimal time for city officials and local residents to prepare for the impending dry rainy season (even though exceptionally strong El Niño conditions had been predicted several months in advance by multiple entities including the National Weather Service Climate Prediction Center and Peru's Estudio Nacional del Fenómeno "El Niño"). Additionally, farmers in the region – many of whom are subsistent – had already made crucial agricultural decisions well before the beginning of the rainy season.

In addition to extended lead times, spatially disaggregated predictions could prove beneficial to several sectors impacted by spatiotemporal precipitation variability. This investigation produces disaggregated predictions with only minimal significant diminishments in skill, which may require further investigation. The governing large-scale climate mechanisms that deliver precipitation to the region more or less act uniformly across this small area of southern Peru, with relatively distinct signals, while station observations may actually be noisier in comparison.

The high correlation between number of wet days and SST in the equatorial Pacific suggests that the ENSO phenomenon not only controls the regional seasonal volume of precipitation, but also the frequency of wet days. While the prediction model of wet days achieves notable skill, the model in general displays a tendency to under-predict the number of wet days (especially

in seasons in which more wet days are observed than dry). Additional prediction skill may be achievable by incorporating local variables such as antecedent soil moisture conditions or low level winds into station-specific models, similar to disaggregation of regional precipitation predictions.

There are several limitations of the framework, including poor performance in predicting above average precipitation conditions and real-time data requirements. Although the region is highly vulnerable to drought conditions, the limited ability of the model to predict above normal conditions could translate into missed economic potential for farmers and mining operations. Improving above normal category prediction could take the form of investigating additional local variables, such as quantifying the orographic impact of the Andes and investigating other small-scale perturbations to the climate system. In

general, though, the prediction framework as developed hinges on readily accessible climate data from the sources used in this study. In some cases, delayed publishing of this data may result in a delayed prediction of JFM precipitation. With some of the ancillary applications, further limitations of the framework include the regional versus local nature of predictions and associated skill and trade-offs with longer prediction lead times.

## 8 Conclusions.

To enhance planning and management for various sectors in southern Peru, a PCR modeling framework is developed to predict JFM seasonal precipitation across the region. Eleven oceanic and atmospheric variables that modulate regional precipitation are identified, with the first four PCs selected for incorporation into the season-ahead prediction model. The PCR model proves skillful, with a clear improvement over climatology and a Niño 3.4 index-based model, and most effective at predicting dry conditions, the state of most interest in this semi-arid region. This points to the evident importance of climatic factors other

than ENSO in modulating regional precipitation. The explored ancillary applications of the model only further demonstrate the potential posed by the framework to provide an array of likely useful information to regional stakeholders.

In this case, the statistical approach explored has produced results that are arguably more skillful than existing methods of precipitation prediction for this region of Peru. Therefore, one may be tempted to draw the conclusion that a statistical approach

of this sort can be applied in a similar fashion at any other location of interest and produce equally skillful results. A conclusion along these lines would be temerarious. Although model frameworks are transferable to other locations, there are no guarantees that one approach will still be superior to another. Furthermore, there is no guarantee that observed increases in skill in one location will translate to expected equivalent increase in skill in another location.

Beyond the work performed in this study, additional avenues for further research include alternative modeling approaches and integration with hydrology and other sectoral/decision-making models. Ideally, stakeholders in the region take advantage of several modeling approaches to make an informed decision regarding water resources management. Other statistical modeling

approaches used in precipitation prediction and the subsequent decision making process could include linear inverse modeling, temporal and spatial pattern tendency modeling, artificial neural networks, Bayesian modeling, etc. When combined with existing capacity to predict and monitor regional precipitation, it may be that stakeholders in southern Peru are able use the results of this and other modeling exercises to shift the existing drought planning paradigm away from reactivity and towards

a more proactive and robust one. The lynchpin of this proactivity is effective and consistent collaboration among ANA, SENAMHI, and other public and private local, regional, and national entities. Projects such as the Peruvian Drought Observatory have served as a starting point for this collaboration; however, the observatory currently offers minimal climate forecast information, and could benefit from the inclusion of such outputs. As drought continues to deleteriously impact water supply and access in southern Peru, season-ahead predictions may become more instrumental in facilitating proactive and

sustainable water management in this semi-arid region of the world.

**Code Availability.**

Should any need arise for the code used in this study, it will be made available upon request by the corresponding author.

**Data Availability.**

Precipitation data used in this study are either purchased from SENAMHI by SPCC or collected by SPCC. This dataset is not

public; however, if needed for additional studies, this dataset may be transferred to appropriate parties. Large-scale climate data come from NOAA and are publicly available for use.

**Appendices.**

None.

**Supplemental Link.**

To be included by Copernicus.

**Team List.**

Eric Mortensen

Shu Wu

Michael Notaro

Stephen Vavrus

Rob Montgomery

José De Piérola

Carlos Sánchez

Paul Block

**Author Contribution.**

Eric Mortensen and Shu Wu contributed to development and evaluation of the prediction model. Michael Notaro and Stephen Vavrus contributed to diagnosis of large-scale climate processes relevant to the project area. Rob Montgomery, José De Piérola, and Carlos Sánchez contributed to contextual understanding of the project area and provided access to relevant documents, literature, datasets, etc. for project team's use. Paul Block contributed to the hydroclimatological analysis, as well as

development and evaluation of the prediction model.

**Competing Interests.**

None.

**Disclaimer.**

To be added later.

**Acknowledgements.**

This study was partially supported by Southern Peru Copper Corporation, including collection of precipitation data across the study region. Additional funding was provided by a seed grant awarded by the Climate, People, and the Environment Program of the Nelson Institute Center for Climate Research at the University of Wisconsin – Madison.

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
