# Peer review of "Regression-based season-ahead drought prediction for southern Peru conditioned on large-scale climate variables"

_Hydrology and Earth System Sciences, 2017_

## Referee Comment (RC1) · Anonymous Referee #1 · 18 May 2017

The paper 'Regression-based season-ahead drought prediction for southern Peru conditioned on large-scale climate variables' by Mortensen et al describes a statistics-based system for seasonal prediction of droughts in Peru. The results are interesting and are relevant to society.

Does the paper address relevant scientific questions within the scope of HESS? yes Does the paper present novel concepts, ideas, tools, or data? not really Are substantial conclusions reached? not really Are the scientific methods and assumptions valid and clearly outlined? yes Are the results sufficient to support the interpretations and conclusions? yes Is the description of experiments and calculations sufficiently complete and precise to allow their reproduction by fellow scientists (traceability of results)? can

be improved. Do the authors give proper credit to related work and clearly indicate their own new/original contribution? Can include more references and cite more previous work on statistical techniques. Does the title clearly reflect the contents of the paper? yes Does the abstract provide a concise and complete summary? yes Is the overall presentation well structured and clear? can be improved with more well-defined data and methods sections with more details. Is the language fluent and precise? yes Are mathematical formulae, symbols, abbreviations, and units correctly defined and used? yes Should any parts of the paper (text, formulae, figures, tables) be clarified, reduced, combined, or eliminated? yes - some repetition. Are the number and quality of references appropriate? could cite some more work with similar techniques or addressing the same type of problems. Is the amount and quality of supplementary material appropriate? NA

Details.

p. 4, L. 1. Monthly precipitation (totals or means?) were derived from presumably daily rain gauge data. It is interesting to look at the number of days per month with precipitation, as the statistical sample of precipitation amounts involves a small (hence greater sampling fluctuations and less well defined mean estimates) in regions with few wet days. To get larger samples, one may use seasonally (3 months) or annually aggregated statistics.

Precipitation may be regarded as having two types of statistical distributions: for dry and for wet days. The dry-day statistics is trivial (zero), whereas the wet-day distribution is often described with the gamma distribution (of exponential for a simple approximation). The classification of the data into 'dry' and 'wet' makes sense because different physical conditions are present when it rains and when it doesn't.

p. 4. L. 2: 'multi-regression' should perhaps be 'multiple regression' or 'multivariate regression'

p. 4. L. 6-11: Please specify if it is version 1 of the reanalysis. Also, the time period

covered and the area selected are important. It is important to have sufficient information so that the analysis can be replicated by others independently. Some of this is discussed further down, but it may be easier for the reader if this is provided in a methods section before the results.

p. 5. L. 1. EOFs often refer to principal components analysis (PCA) of gridded data, weighted by the gridbox area. PCA is mathematically the same thing, but a term used more generally than EOFs. However, this is a matter of taste.

p. 5. L. 8. How much of the variance do the subsequent modes capture, and presumably the second order suggests a bi-pole type pattern? There is no need to show this, but perhaps worth describing its character. It is interesting that the leading PC so closely reproduces the station (not area?) mean precipitation,. What does that suggest? That the precipitation is dominated by large-scale climatic phenomena (at least aggregated over 3 months) and that other modes are essentially regional perturbations from the large-scale precipitation? I think it may be worth commenting these aspects, but perhaps later in the discussion.

Unless the precipitation has been gridded to onto a regular mesh, the term 'area average' should be replaced with 'station average'.

p.6, L. 10. Perhaps the higher modes of the PCA shows the orographic effects.

P. 8. figure 6: the scatter suggests that more than one factor affects the precipitation, but there is also a discernible anti-correlation between Nino3.4 and the precipitation. An ordinary linear regression can quantify the relationship (and associate a p-value), as can the correlation coefficient. This can be repeated with a subset of the data where the three outliers are excluded to estimate how exceptional they were.

p. 11, L11-16. subtracting 9 driest season during El nino Years from the 9 wettest from La Nina years is bound to produce an ENSO signal by design of the analysis. This paragraph seems to be repeated on p. 12, L. 1-5.

p. 13., L. 10.14. Is is evident that the PCA applied to area means according to Table 1., but this should also be stated more clearly in the main text. Also, the text should explain how different units are handled - was the PCA applied to standardised indices?

p. 14, L. 16-17. It is not clear how hindcasts were generated from residuals from cross-validation. Please elaborate.

p. 19, L.6-8. From a sample of test results, one will expect som to score high from pure chance. If there are 100 stations, one would expect 5 to score above the 95% confidence level by chance - this is how the confidence interval is defined.

split discussions and conclusions into two sections. The conclusions should be brief and repeat the main findings.

---

## Author Comment (AC1) · 15 Aug 2017

Response to Anonymous Referee #1

We thank Referee #1 for carefully reviewing our manuscript and providing critical and valuable comments. Please find below responses to each point raised by the Referee.

**Comment**: p. 4. L. 1. *Monthly precipitation (totals or means?) were derived from presumably daily rain gauge data. It is interesting to look at the number of days per month with precipitation, as the statistical sample of precipitation amounts involves a small (hence greater sampling fluctuations and less well defined mean estimates) in regions with few wet days. To get larger samples, one may use seasonally (3 months) or annually aggregated statistics. Precipitation may be regarded as having two types of statistical distributions: for dry and for wet days. The dry-day statistics is trivial (zero), whereas the wet-day distribution is often described with the gamma distribution (of exponential for a simple approximation). The classification of the data into 'dry' and 'wet' makes sense because different physical conditions are present when it rains and when it doesn't.*

**Response:** We thank the Referee for this excellent suggestion. Indeed, we initially considered this exact construct to evaluate the predictability of the frequency of precipitation events or wet/dry days, but omitted due to extra manuscript length. However, we have determined that it could be added with minimal extension to the manuscript and are pleased to do so as suggested.

First, for clarification, to this point we have used monthly and seasonal total precipitation, not daily averages, even though the original gauge data is at the daily scale, as correctly presumed by the Referee. (It should be noted, however, that for this study daily data was only readily available from the six stations maintained by SPCC).

We have framed this new piece as complementary to the seasonal total prediction approach as opposed to an entirely unique undertaking. For this auxiliary analysis, we identify the number of wet days as our predictand. As in the original analysis, we use principal component spatial correlation mapping, composite mapping, and global wavelet analysis in an attempt to identify potential predictors of number of wet days.

Interestingly, approximately 85% of the variance experienced in the number of wet days for all stations was captured by the first principal component (PC) of the data. Spatial correlation of the first PC with global SST revealed high correlations within the region of the Pacific typically associated with ENSO, suggesting that the ENSO phenomenon not only controls the seasonal volume of precipitation that reaches the region, but also the number of days during which it falls. No additional regions or climate variables (e.g. sea level pressure, geopotential height, etc.) were identified using this method. Further, composite mapping and global wavelet analysis yielded no additional potential predictors for incorporation into the prediction model. Therefore, the wet-day prediction model uses only the OND Niño 3.4 season-ahead ENSO index as a direct predictor to produce a deterministic prediction of number of wet days in any given JFM season.

New text is added throughout the manuscript in reference to this auxiliary analysis. Specifically, the following will be included in Section 7 of the revised manuscript (now retitled as Additional Prediction Modifications, with three subsections: Extended Lead Time; Spatial Disaggregation of Regional Predictions; and Prediction of Wet/Dry Days):

> "Although predictions of JFM seasonal precipitation totals may be useful for a variety of stakeholders throughout the region, some may prefer additional detailed information such as the frequency of precipitation events expected across a given rainy season. The number of wet or dry days and the intensity of precipitation events can have widespread and serious agronomic/phenologic (Robertson et al., 2008) and infiltration/runoff implications (Mandal and Nandi, 2017). Such information may also be informative to condition stochastic weather simulators for a wide range of hydrologic or agricultural models (Robertson et al., 2006). To evaluate seasonal statistics of wet/dry day frequency for southern Peru, six of the 29 stations having daily data are analyzed. Analogous to the seasonal total precipitation prediction modeling approach, spatial correlation mapping, composite mapping, and global wavelet analysis are all utilized to identify potential predictors describing the expected number of wets days across the JFM season.
>
> Interestingly, approximately 85% of the variance in the number of wet days for all six stations was captured by the first principal component of the data. High correlations are observed between this first principal

component and SST within the region typically associated with ENSO. No additional regions or climate variables (e.g. sea level pressure, geopotential height, etc.) were identified using this method. Further, composite mapping and global wavelet analysis yielded no additional potential predictors for incorporation into the prediction model. Therefore, the wet-day prediction model uses only the OND Niño 3.4 season-ahead ENSO index as a direct predictor to produce a deterministic prediction of number of wet days in any given JFM season. Using the same cross-validation method described previously, the number of wet days per season is predicted for each station.

In addition to correlation coefficients between the predicted and observed number of wet days, the average prediction error for above-average and below-average years is reported for each station. Station-specific statistics are listed in Table 4.

**Table 4: Correlation values and average absolute errors for predictions of wet days at each station.**

| Station (with average number of wet days) | Correlation value between prediction and observation | Average absolute error in years with above average number of wet days | Average absolute error in years with below average number of wet days |
|---|---|---|---|
| 1 (25 days) | -0.50 | 10 days | 9 days |
| 2 (36 days) | -0.53 | 11 days | 9 days |
| 3 (51 days) | -0.48 | 11 days | 10 days |
| 4 (54 days) | -0.39 | 12 days | 12 days |
| 5 (54 days) | -0.53 | 8 days | 9 days |
| 6 (33 days) | -0.59 | 10 days | 9 days |

Correlations between predictions and observations range from -0.39 to -0.59, with the model performing slightly better in predicting wet days in drier years. In general, however, the simple linear model displays an average absolute error ranging between 8 and 12 days.

To consider overall model performance, a hit miss metric quantifies skill in the two previously introduced categories of years – years with above average number of wet days and years with below average number of wet days (Table 9).

**Table 9: Hit miss metric for model predictions of years with above and below average numbers of wet days.**

| | | Predicted conditions | |
|---|---|---|---|
| | | W | D |
| Observed conditions | W | **105** | 41 |
| | D | 43 | **103** |

Above average number of wet days (W) and below average number of wet days (D)

Overall, the model correctly predicts whether a given season will have an above or below average number of wet days over incorrectly predicting, with an accuracy of ~72%. The model, however, has a notable bias towards over-predicting near normal conditions (Fig. 13).

[Figure]

**Figure 13: Observed number of wet days compared to predictions."**

In addition, we propose adding the following paragraph to Section 8 (Discussion) of the manuscript:

> "The high correlation observed between number of wet days and SST in the equatorial Pacific suggests that the ENSO phenomenon not only controls the regional seasonal volume of precipitation, but also the frequency of wet days. While the prediction model of wet days achieves notable skill, the model in general displays a tendency to under-predict the number of wet days (especially in seasons in which more wet days were observed than dry). More benefit may be gained in using this model at locations with more wet days (such as stations 3 and 4) as opposed to drier stations. Additional prediction skill may be achieved by incorporating local variables such as antecedent soil moisture conditions or low level winds into station-specific models, as is the case with the disaggregation of regional precipitation predictions."

In addition to precipitation event frequency, the Referee also comments on precipitation distribution for wet days. Indeed, this is legitimate; however, we have opted not to include this additional analysis here as we are presently less focused on understanding daily precipitation statistics or fitting stochastic weather generators. That said, it does lay the groundwork for others that may be focused on such applications.

**Comment:** p. 4. L. 6-11. *Please specify if it is version 1 of the reanalysis. Also, the time period covered and the area selected are important. It is important to have sufficient information so that the analysis can be replicated by others independently. Some of this is discussed further down, but it may be easier for the reader if this is provided in a methods section before the results.*

**Response:** We acknowledge that we did not initially make this clear and have thus updated the manuscript to explicitly state that version 1 of the reanalysis data is used in this study. Additionally, we have clearly noted the period covered by this dataset (1948 to present). To facilitate introducing this information earlier, as suggested, the geographic areas selected and listed in Table 1 of the paper are now referenced in Section 2 (Data Descrpition).

**Comment:** p. 5. L. 8. *How much of the variance do the subsequent modes capture, and presumably the second order suggests a bi-pole type pattern? There is no need to show this, but perhaps worth describing its character. It is interesting that the leading PC so closely reproduces the station (not area?) mean precipitation. What does that suggest? That the precipitation is dominated by large-scale climatic phenomena (at least aggregated over 3 months) and that other modes are essentially regional perturbations from the large-scale precipitation? I think it may be worth commenting these aspects, but perhaps later in the discussion.*

**Response:** The Referee highlights a good point, and we agree that it should be further clarified and discussed. The first mode clearly explains the majority of the variance in data (50%), and the second mode captures an additional ~20% of variance; however, the third drops to ~5%. Only these three modes are investigated in this study, for a cumulative total of 75% of variance captured (Fig. 1). The manuscript has been revised to state the variance explained by each mode considered. Indeed, as suggested by the Referee, the second PC suggests a dipole pattern; this description has also been added to the revised manuscript.

[Figure]

**Figure 1: From left to right, spatial pattern produced by the first, second, and third modes of the PCA performed on regional precipitation, explaining ~50%, ~20%, and ~5% of the variance, respectively.**

As referenced in the manuscript, Eklundh and Pilesjö (1990) postulated that high correlations between the first EOF of gridded precipitation and area averaged precipitation may suggest the presence of a large-scale climatic phenomena acting homogenously on regional precipitation. Additional studies that support this notion include Ogallo (1980), Mallants and Feyen (1990), Bisetegne et al. (1986). While we use a station average precipitation time series (and not area averaged, as noted by the Referee), the high correlation coefficient between this time series and the first PC of the original set of data may still be interpreted as a widespread homogenous influence on regional precipitation by a large-scale climatic phenomenon. With a correlation between the first principal component of regional JFM precipitation and JFM Niño 3.4 of -0.52, it is likely that this PC describes the modulation stemming from ENSO. As the Referee mentions, the subsequent PCs likely describe regional and local perturbations.

The additional studies mentioned in this response are now referenced in the manuscript.

Additionally, we will add text to Section 8 (Discussion) interpreting the character of each of the PCs considered.

**Comment:** p. 6. L. 10. *Perhaps the higher modes of the PCA shows the orographic effects.*

**Response:** The Referee is right that the orographic effects may be evident in various spatial patterns (EOFs), and particularly likely in the higher order modes given that they are more likely associated with regional and local influences. Although this characteristic is not fully explored here, we have added a sentence to the manuscript in Section 3 (Southern Peru Rainy Season and Large-scale Climate Influences) acknowledging that this may indeed be the case:

> "While the first PC likely illustrates ENSO's influence on regional precipitation, it is possible that higher order modes may describe other climatic and topographic forcings such as large-scale climatic phenomena or observed orographic effects."

**Comment:** p. 8. *Fig. 6: the scatter suggests that more than one factor affects the precipitation, but there is also a discernible anti-correlation between Nino3.4 and the precipitation. An ordinary linear regression can quantify the relationship (and associate a p-value), as can the correlation coefficient. This can be repeated with a subset of the data where the three outliers are excluded to estimate how exceptional they were.*

**Response:** The Referee is right in assessing that Nino3.4 is highly influential but also not the only relevant signal in describing precipitation variability. The correlation between JFM precipitation and JFM Niño 3.4 (i.e., concurrent conditions) is -0.57 (p-value = 0.000013). Due to the autoregressive nature of SST in general, Niño 3.4 maintains a relatively high correlation with JFM precipitation for the prior season as well. This has been added to the text and Fig. 6. If the three outlier years (1973, 1990, and 2014) are removed, as suggested by the Referee, this produces a stronger correlation coefficient of -0.66 (p-value = 0.000009), supporting the two-fold notion of ENSO's influence on seasonal precipitation as well as the presence of additional climatic factors that modulate the region's precipitation. This evaluation has also been added to the text immediately following to estimate the importance/influence of these three highly anomalous years, and further underscores the importance of creating a model conditioned on several potential predictors rather than just an ENSO index.

**Comment:** p. 11. L. 11-16. *Subtracting 9 driest season during El Niño years from the 9 wettest from La Niña years is bound to produce an ENSO signal by design of the analysis. This paragraph seems to be repeated on p. 12. L. 1-5.*

**Response:** The Referee highlights a valid point. The example composite map, as framed in the manuscript, produces an ENSO signal because only El Niño years and La Niña years are selected for compositing. Thus, this example only marginally supports the points emphasized in this section. Thus, we have removed this figure and text and substituted instead with analysis demonstrating the presence of ENSO through compositing simply the driest and wettest years, regardless of ENSO phase and strength. Accordingly, the manuscript has been revised as follows, including a new Figure 8:

> "Composite maps illustrate climate conditions for a single period or subset of periods, and may be especially useful for understanding forcing mechanisms in anomalous periods. For example, OND SST for the nine subsequent driest JFM seasons on record for southern Peru subtracted from OND SST for the nine subsequent wettest JFM seasons on record for southern Peru produce large positive anomalies in the equatorial Pacific Ocean. This composite map (Fig. 8) further indicates the potential importance of ENSO in explaining JFM precipitation variability in the study region.

[Figure]

**Figure 8: Composite SST OND conditions of nine subsequent driest JFM seasons (1983, 1992, 1990, 1966, 2014, 2016, 1980, 2010, and 1979) and subtracted SST OND conditions of nine subsequent wettest JFM seasons (2001, 1984, 1986, 1972, 1973, 1999, 2012, 1974, and 1997)."**

Additionally, any unintended repetition with regard to composite mapping has been removed from the manuscript.

**Comment:** p. 13. L. 10-14. *It is evident that the PCA applied to area means according to Table 1, but this should also be stated more clearly in the main text. Also, the text should explain how different units are handled - was the PCA applied to standardized indices?*

**Response:** Thanks for pointing out the need for better clarity. The text has been revised to include reference to the fact that PCA is applied to area mean values of the selected potential predictors. Also, all predictor values are standardized in the PCA process so as not to cause any unintended preference. The text has been appropriately revised.

**Comment:** p. 14. L. 16-17. *It is not clear how hindcasts were generated from residuals from cross-validation. Please elaborate.*

**Response:** We agree that this may not have come across clearly in the original manuscript. Hindcasts for precipitation prediction are performed using principal component regression in a drop-one cross-validated mode. This includes – for each year of the hindcast – dropping the predictor data (Z) from the year being hindcasted, forming new PCs (and EOFs) conditioned on the remaining years, and fit to observations using multiple regression, providing an intercept coefficient, regression coefficients, and error term. The predictor data (Z) from the year dropped are then projected onto the EOFs to provide PCs for the dropped year. Finally, these PCs are multiplied by the appropriate regression coefficients and added to the intercept coefficient to provide a deterministic precipitation prediction for the dropped year. This is repeated for each year.

To create ensemble hindcasts, error terms from all years are assembled and a distribution is fit (using a kernel density estimator; the distribution is approximately Gaussian). For each hindcast year, 1000 random draws from the distribution are added to the deterministic precipitation prediction to form an ensemble.

The manuscript has been revised accordingly to clarify this process.

**Comment:** p. 19. L. 6-8. *From a sample of test results, one will expect some to score high from pure chance. If there are 100 stations, one would expect 5 to score above the 95% confidence level by chance - this is how the confidence interval is defined.*

**Response:** The Referee makes a valid point. The manuscript text has been modified appropriately to reflect this, as follows:

> "None of the stations experience statistically significant decreases in correlation as a result of disaggregation. For two of the stations, correlation values between station-level predictions and station-level observations increase (statistically significant at the 95% confidence interval) as compared with correlation values between regional-level predictions and station-level observations. (However, it should be noted that there is a 5% probability that this happens by chance.) The remaining stations exhibit no statistically significant change in correlation as a result of station-level scaling."

**Minor Comments (all accepted and corrected in the manuscript)**

p. 4. L. 2. *'multi-regression' should perhaps be 'multiple regression' or 'multivariate regression'*

p. 5. L. 1. *EOFs often refer to principal components analysis (PCA) of gridded data, weighted by the gridbox area. PCA is mathematically the same thing, but a term used more generally than EOFs. However, this is a matter of taste.*

p. 5. L. 8. *Unless the precipitation has been gridded to onto a regular mesh, the term 'area average' should be replaced with 'station average'.*

p. 19 L. 12. *Split discussions and conclusions into two sections. The conclusions should be brief and repeat the main findings.*

**References**

Bisetegne, D., Ogallo, L., and Ininda, J.: Rainfall characteristics in Ethiopia, Technical Conference on Meteorological Research in Eastern and Southern Africa, Nairobi, Kenya, 1986.

Mallants, D., and Feyen, J.: Defining Homogenous Precipitation Regions by Means of Principal Component Analysis, Journal of Applied Meteorology, 29, 892-901, 1990.

Mandal, A., and Nandi, A.: Hydrological Modeling to Estimate Runoff and Infiltration in Southeastern Appalachian Debris Flow Complex., 3rd North American Symposium on Landslides, Roanoke, Virginia, USA, 2017.

Ogallo, L.: Regional classification of East African rainfall stations into homogeneous groups using the method of principal component analysis, as in Ikeda, S. et a1. (eds), Statistical Climatology. Developments in Atmospheric Sciences, 13. 225-266, Elsevier, Amsterdam, 1980.

Robertson, A., Kirshner, S., Smyth, P., Charles, S., and Bates, B.: Subseasonal-to-inderdecadal variability of the Australian monsoon over North Queensland, Quarterly Journal of the Royal Meteorological Society, 132, 519-542, 2006.

Robertson, A., Moron, V., and Swarinoto, Y.: Seasonal predictability of daily rainfall statistics over Indramayu district, Indonesia, International Journal of Climatology, 29, 1449-1462, 2008.

---

## Referee Comment (RC2) · S. Harrigan (Referee) · 1 Sep 2017

**Review of Mortensen et al. (2017) in HESSD**

**A.) General Comments**

Mortensen et al. (2017) presents a new statistically-based seasonal forecast model for prediction of the three month wet season (January-March) in Southern Peru, with a focus on drought. The statistical model is based on principal component regression (PCR) using 11 large-scale climate predictors derived from SST or SLP fields, such as El Niño 3.4. There is good justification for using PCR as the authors highlight it removes the well-known issue of multicollinearity suffered by traditional multiple-linear regression models. The model building and skill evaluation is robust. A long 51-year hindcast period was used, cross-validation for both model building and skill evaluation exercised, and multiple evaluation metrics considered such as correlation, RPSS, and tercile contingency tables as well as for a more extreme precipitation drought category. The hindcast results showed low to moderate skill improvements over climatology as well as a simpler ENSO based model, highlighting a more complex climate-precipitation link than often assumed thus the added benefit of considering a wider set of predictor variables. The model developed has direct practical application in Southern Peru, a region where the availability of seasonal forecasts is limited.

I found this paper interesting from a seasonal forecast and drought prediction perspective given the limited available alternatives in Southern Peru and indeed in many other regions of the world at this lead time and think it deserves publication in the HESS special issue on 'Sub-seasonal to seasonal hydrological forecasting'. There are however several aspects of the paper that need revised before publication. These mainly concern the lack of international literature cited throughout on seasonal/statistical forecasting, the non-standard HESS layout, and a few other minor comments/suggestions that I've outlined below together with some suggestions for improvement.

**B.) Specific Comments**

1. Throughout the text, including the title, the term 'drought' is used with many statements making the connection to a "hydrologic extreme" (Pg1; L17) and for example "the city's water supplies were reduced" (Pg2; L13-14) etc. It took me until the last line of the introduction (excluding the abstract) to realise it was prediction of meteorological drought (precipitation) rather than hydrological drought that was being pursued. Clearly improving prediction of meteorological drought is still valid, but precipitation deficit does not necessarily propagate to a soil moisture, streamflow, and/or groundwater drought, which are more societally relevant. Often there are more complex processes at play, including temperature/evapotranspiration feedbacks. See Van Loon (2015) for more detail. This should be acknowledged and please be more explicit about the focus on meteorological drought throughout. I think if you are more explicit within the abstract and main text I would not be pedantic about asking to change the title, as it is a good title.

2. I am not convinced the layout of the paper fits reasonably with traditional HESS style. Nor does it help convey the story of the paper as good as it could. I acknowledge the story here is a little more complex and awkward to fit into exactly the traditional style, for example I think it is appropriate to have Sect. 3 as 'Southern Peru Rainy Season and Large-scale Climate Influences', it works well. However:

   a) The introduction does not do the paper justice as it fails to clearly establish core research aims/objectives/questions. The fundamental finding of the paper is that the newly created PCR model was found to be more skilful that a climatological forecast AND a simpler Niño 3.4

index-based model forecast, especially for dryer conditions – Is this not the foundation of your research question(s)? If so, needs to be in the Introduction.

 **b)** Even though drought prediction remains largely unexplored in Peru, there is no background nor reference to the international literature on general seasonal forecasting methods in the introduction section, nor previous work done internationally on statistical forecasting. For example, what is the justification for selecting a statistical forecasting approach over others (i.e. lack of climate/hydrological modelling, limited hydrological data,...)?

 **c)** Methods and Results are scattered over several sections. Reforming methods into a 'Section 4 Methods' and 'Section 5 Results' together with the use of sub-headings would help. I especially found it frustrating to have Sect. 7 after the results section. Surely this should not be hard to have the results divided by sub-heading for 'Sect. 5.1 Season-ahead' and 'Sect. 5.2 Extended Lead Time and Spatial Disaggregation of Regional predictions', or something similar.

 **d)** The 'Summary and Discussion' section does an excellent job of outlining the practical implications of the work. However, it does not discus results in light of the international forecasting literature. Is the degree of increased skill on par with other areas/forecasting approaches?

**3.** Pg3; L5: The elevation range in Peru is substantial. It would therefore be beneficial for an international audience to provide the mean or median and the range of elevation for the 29 precipitation stations.

**4.** Pg 4; L3: what is the average correlation and is the method used Pearson?

**5.** Pg 4; 21: Agree a focus on JFM is justifiable. To help convince the reader that forecasting the wettest season is relevant to water resources/drought perhaps worth mentioning that it is during the wet season that reservoir/aquifer stores are replenished for use during dryer summer months. Being able to skilfully forecast anomalously low precipitation for the wet season is indeed valuable.

**6.** Pg4; Fig. 2 caption: Add "…using data from 29 precipitation stations in Sect. 2".

**7.** Pg5; L2: Referring to both EOF and PCA throughout. Stick with one to avoid confusion.

**8.** Pg5; L7: The Eklundh and Pilesjö (1990) reference is for Ethiopia. Should reference go after "homogeneity" instead?

**9.** Pg5 & 6; Fig. 3 & 4: It was not clear to me why PCA was used for the observed JFM precipitation totals? What purpose does it serve if the main target for the PCR model is for areal averaged precipitation totals anyway? Also, it is not stated what the physical interpretation of PC2 and PC3 are in this context (i.e. Pg9; L26 & Pg10; L9). As it stands Fig. 3 does not really add anything. It is too difficult to see any difference the size of the red dots. Perhaps adding a scale and/or some gradual colour scale would help? Could you include what elevation threshold the topographic shading represents?

**10.** Pg6; L12-13: Am I correct in thinking none of the precipitation stations used here are within the rain shadow?

**11.** Pg6; 20: Are strong El Niño's associated with only low precipitation or are they associated with actual societally impactful droughts such as problems with agriculture, water supply etc.?

**12.** Pg 7; Fig. 5: Add units of SST anomalies (i.e. °C)? You could improve plot by adding two horizontal lines to represent El Niño/La Niña thresholds (i.e., $\pm 0.5$°C). Also define that you are using

Pearson's correlation coefficient (I presume?) in the first instance (in the text) and define symbol as $r$. Then use $r$ in every instance throughout the paper for clarity. I note this is done in some places and not in others (e.g. Pg9; L30).

13. Pg8; 5-8: Very short paragraph, better added to the previous one?

14. Pg9; 21: Change "previously identified" with "identified in Sect. 3".

15. Pg10 & 11; Fig. 7 & 8: Worth adding a dot/circle to mark study region on the maps for an international audience?

16. Pg10; Fig. 7: Why using the first PC of regional JFM precipitation instead of just the area-averaged JFM precipitation total? I do note these are very similar and map would look the same.

17. Pg11; L1-5: Delete section as repeated from Pg10.

18. Pg12; L3: "the JFM precipitation series"… but which one? First PC or observed totals?

19. Pg13; Table 1 & L1-4: The use of the asterisks is a little confusing. When I first looked at table 1 I presumed the asterisks was for statistically significant correlations. But are these instead those that are NOT correlated with JFM precipitation? Although you say that all are significant with at least one of the first three PCs. I can see here how perhaps the use of the first three PCs is useful but the reader is left with a bit of a jump to understand this without understanding what PC2 and PC3 represent. Could adding three additional columns to the right hand side of Table 1 for PC1, PC2, and PC3 help with this, then have the asterisks marking any value with statistically significant correlations. This allows the reader to see that perhaps one climate variable is correlated with all four precipitation series, or just e.g. PC3?

20. Pg14; L18-19: Not clear how the ensemble in Fig. 10 was created. More detail needed here. Also, how many ensemble members etc.?

21. Pg15;L6: A few issues with this sentence. Suggest changing to something like: "An RPSS value less than zero signifies no forecast skill over the reference climatology forecast (i.e. it is ….), a value equal to zero for when the forecast is only as skilful as climatology, and values greater than zero represents a skilful forecast. A value of one represents a perfect forecast".

22. Pg15; L9-14: Need more details about the use of 3x3 contingency tables, you might find the Svensson (2016) paper (and references therein) useful for this and as an example of statistical seasonal forecasting more generally. Also, more definition of what is meant by "extremely dry conditions". I know this is mentioned in the results, but it should be here that the methods details are given.

23. Pg15; L16-18: Which combinations of the 11 predictors in Table 1 made it into the final PCR model? I know PCA was used, but can weight be given to original 11 predictor variables? For example, can I tell how important, if at all, Niño 3.4 is to the final model?

24. Pg16; L5-7: The main message I get from Table 2 is that Near normal and Below normal precipitation is good, but it is the above normal that drags the hit rate to 51%.

**25.** Pg17; L21-27: This is a key conclusion. That the new model is more skilful than both climatology and a simpler ENSO only model is central to the paper. A moderate skill improvement at a seasonal level is still valuable. Also interesting that RPSS is negative for Niño 3.4 only model, why do you think this is? Overall, re-framing the introduction to include this science question at the start would strengthen the paper. This finding is lost as it is, so you need to place this moderate but important improvement in light of the ongoing work in this active area of research internationally, and not only the practical uses of the forecast in S. Peru.

**26.** Pg17; Sect. 7: I like the idea of extended lead time analysis, but the technical details should be first outlined in the proposed 'Methods' section and results presented under a sub-heading within the 'Results' section.

**27.** Pg18; Fig. 12: I'm missing how you are going from regional level to station level here. The extended lead time is good, but the spatial disaggregation is the weakest part of the analysis at present.

**28.** Pg19; L6-9: What statistical test is used to determine if the difference between regional and station correlation values is statistically significant or not?

**29.** Pg19; Sect. 8: There is no discussion of the key limitations of the forecasting method/model (e.g. poor for above average precipitation). It would be good here to offer some suggested avenues for further research to overcome such methodological limitations.

**30.** I like the final paragraph on Pg 20 as it highlights well the practical importance of seasonal forecasting using climate information in a region where none is currently available.

**C.) Technical Corrections**

1. The paper is generally well written, but in places language is a bit colloquial. E.g.
   a) Pg1; L28: Change "vary drastically" to e.g. "vary considerably"
   b) Pg2; L19: Change "wreaked havoc" to e.g. "was particularly severe"
   c) Pg2; L26: Change "The dire" to e.g. "The societally challenging"

2. Pg3; Fig. 1 caption: Do white circles not represent SPCC stations, and blue the SENAMHI?
3. Pg6; 11: Add a comma in (Garreaud, 1999)
4. Pg9; L3: Change "hydrometeorologic" to "hydrometeorological"? Perhaps this term is used in the US?
5. Pg13; Table 1: Add space between 'Time frame'
6. Pg18; L9: Add full stop after "…etc.)".
7. Pg1; 19: Add "regional" and "totals" in front of and after "January-March precipitation", respectively?

**References**

Svensson, C.: Seasonal river flow forecasts for the United Kingdom using persistence and historical analogues, Hydrol. Sci. J., 61(1), 19–35, doi:10.1080/02626667.2014.992788, 2016.

Van Loon, A. F.: Hydrological drought explained, Wiley Interdiscip. Rev. Water, 2(4), 359–392, doi:10.1002/wat2.1085, 2015.

---

## Author Comment (AC2) · 6 Oct 2017

Response to Referee #2

We thank Dr. Harrigan for carefully reviewing our manuscript and providing critical and valuable comments. Please find below responses to each point raised.

**Comment:** *Throughout the text, including the title, the term 'drought' is used with many statements making the connection to a "hydrologic extreme" (*p. 1. L. 17.*) and for example "the city's water supplies were reduced" (*p. 2. L. 13-14.*) etc. It took me until the last line of the introduction (excluding the abstract) to realize it was prediction of meteorological drought (precipitation) rather than hydrological drought that was being pursued. Clearly improving prediction of meteorological drought is still valid, but precipitation deficit does not necessarily propagate to a soil moisture, streamflow, and/or groundwater drought, which are more societally relevant. Often there are more complex processes at play, including temperature/evapotranspiration feedbacks. See Van Loon (2015) for more detail. This should be acknowledged and please be more explicit about the focus on meteorological drought throughout. I think if you are more explicit within the abstract and main text I would not be pedantic about asking to change the title, as it is a good title.*

**Response:** We apologize for not being clearer in our manuscript. Our goal is to create a predictive model to prognosticate meteorological drought conditions, and not necessarily consequential hydrologic drought. While these two topics are intrinsically related, we fully agree they are not interchangeable. The text of the manuscript has been changed such that meteorological drought is more explicitly stated as the focus of the research, with references to the tangential potential of hydrologic drought given meteorological drought conditions.

**Comment:** *The introduction does not do the paper justice as it fails to clearly establish core research aims/objectives/questions. The fundamental finding of the paper is that the newly created PCR model was found to be more skillful than a climatological forecast AND a simpler Niño 3.4 index-based model forecast, especially for dryer conditions – Is this not the foundation of your research question(s)? If so, needs to be in the Introduction.*

**Response:** We appreciate the opportunity to clarify our objectives in the Introduction, and propose adding the following paragraph to the manuscript:

> "In this paper, we develop a season-ahead principal component regression (PCR) model to predict seasonal precipitation totals. This PCR model draws on a pool of large-scale climate variables that influence southern Peru precipitation through ocean-atmosphere teleconnections. The model is evaluated against climatology and simpler Niño index-based models to understand if the inclusion of several predictors leads to more skillful prediction, particularly for dry years in this drought-sensitive region."

**Comment:** *Even though drought prediction remains largely unexplored in Peru, there is no background nor reference to the international literature on general seasonal forecasting methods in the introduction section, nor previous work done internationally on statistical forecasting. For example, what is the justification for selecting a statistical forecasting approach over others (i.e. lack of climate/hydrological modelling, limited hydrological data...)?*

**Response:** Statistical forecasts have been developed and evaluated for many applications globally, although more effort is still focused on the application of dynamical model predictions. There are numerous advantages for selecting statistical models for season-ahead precipitation prediction over other methods involving global atmospheric general circulation models, most notably reviewed by Xu (1999). These include GCMs inability to represent sub-grid features and dynamics, vertical level mismatches between GCMs ability and hydrology needs (atmospheric vs surface), and discrepancies in the importance placed on variables used in dynamical models. Essentially, dynamical models are exceptional tools for macroscale climate modeling, but struggle in scales similar to that of our project area. Additionally, the complex topography of the region complicates the use of GCMs for regional predictions. Although we are focused on meteorological drought and not necessarily hydrologic drought, the interconnectedness of these

two types of drought necessitates a methodology that can be justified if meteorological predictions were to be used for hydrologic application. For this case, the merits of statistical modeling outweigh those of dynamical modeling.

We have revised the manuscript to include these advantages and reference Xu's review. Although the full application of statistical forecast models is clearly too wide to synthesize, as we hope the Referee agrees, we have attempted to justify why the selection of a statistical modeling approach is valid in this case.

**Comment:** *Methods and Results are scattered over several sections. Reforming methods into a 'Section 4 Methods' and 'Section 5 Results' together with the use of sub-headings would help. I especially found it frustrating to have Sect. 7 after the results section. Surely this should not be hard to have the results divided by sub-heading for 'Sect. 5.1 Season-ahead' and 'Sect. 5.2 Extended Lead Time and Spatial Disaggregation of Regional predictions', or something similar.*

**Response:** We agree with this recommended restructuring. Our original intention was for Section 7 to serve as a supplement to the focus of the manuscript, however we fully acknowledge that integrating the topics of extended lead-time, spatial disaggregation, and prediction of wet/dry days (as recommended by Referee #1) throughout the manuscript is warranted and improves the flow. Changes to the manuscript include moving the technical details of these additional applications into the Introduction, Methods and Results (newly named sections as suggested) as appropriate. The authors are appreciative of the Referee's suggestion.

**Comment:** *The 'Summary and Discussion' section does an excellent job of outlining the practical implications of the work. However, it does not discuss results in light of the international forecasting literature. Is the degree of increased skill on par with other areas/forecasting approaches?*

**Response:** The Referee raises an interesting point. The skill achieved in this study surpasses that of existing approaches – namely dynamical models and simple Nino-based index models – as we hope has been illustrated and presented clearly. The challenge of course is putting this in the context of progress made elsewhere, and whether the improvements presented here represent a moderate or sizeable step forward. To be honest, comparing across climatologically diverse case studies is nontrivial. We can argue with some assurance that we have demonstrated improvement in comparison with other (existing) model structures for the same set of circumstances (i.e. same set of observations), however this may not hold true for a similar experiment in another location. Model performance is very site specific in our experience, including the "best" modeling approach.

We may argue that statistical approaches "typically" outperform dynamical models in predicting local precipitation, but this is not universal, and there are certainly a number of caveats. To this end, we have added a paragraph that discusses this idea and thank the Referee for highlighting this point:

> "In this case, the statistical approach explored has produced results that are arguably more skillful than existing methods of precipitation prediction for this region of Peru. Therefore, one may be tempted to draw the conclusion that a statistical approach of this sort can be applied in a similar fashion at any other location of interest and produce equally skillful results. A conclusion along these lines would be temerarious. Although model frameworks are transferable to other locations, there are no guarantees that one approach will still be superior to another. Furthermore, there is no guarantee that observed increases in skill in one location will translate to expected equivalent increase in skill in another location."

**Comment:** p. 3. Fig. 1: *Do white circles not represent SPCC stations, and blue the SENAMHI?*

**Response:** We thank the Referee for noticing this error. Indeed, the six white circles represent SPCC stations, and the blue circles represent SENAMHI stations. The caption of Fig. 1 has been revised accordingly.

**Comment:** p. 3. L. 5. *The elevation range in Peru is substantial. It would therefore be beneficial for an international audience to provide the mean or median and the range of elevation for the 29 precipitation stations.*

**Response:** The topography of the region is noteworthy. Section 2 of the manuscript has been revised as follows to provide this context:

> "The topography of the region is noteworthy. While the 29 stations considered in the study cover an elevation range from 3,100 m to 4,600 m (Fig. 2, mean elevation 3870 m), this portion of southern Peru ranges from sea level at the Pacific Ocean to over 6,000 m in the high Andes.

[Figure]

**Figure 2: Elevations of all 29 stations included in the study. Bars are numbered and colored in accordance with Fig. 1, with white bars representing SPCC stations and blue bars representing SENAMHI stations."**

The station numbers referenced in Fig. 2 have been added to Fig. 1 to easily identify any station's location within the region and quantify its elevation. Figure numbers throughout the remainder of the manuscript have been revised accordingly.

**Comment:** p. 4. L. 3. *What is the average correlation and is the method used Pearson?*

**Response:** The average Pearson's correlation coefficient for the missing points is 0.92, implying that this interpolated data is reasonably representative of actual conditions not captured by the data. We have also clarified in the text that we are using Pearson correlation throughout.

**Comment:** p. 4. L. 21. *Agree a focus on JFM is justifiable. To help convince the reader that forecasting the wettest season is relevant to water resources/drought perhaps worth mentioning that it is during the wet season that reservoir/aquifer stores are replenished for use during dryer summer months. Being able to skillfully forecast anomalously low precipitation for the wet season is indeed valuable.*

**Response:** The following passage has been added to the manuscript to reflect the relevance of seasonal precipitation to regional hydrology and water resources management:

> "JFM precipitation represents, on average, more than two-thirds of annual precipitation for the region, with some locations receiving up to 85% of annual precipitation during the three-month period. This precipitation

is crucial to the region's economic activities and environmental stability. During the rainy season, for example, surface reservoirs and underground aquifers are replenished for multi-sectoral water resource use during the dry conditions that characterize the rest of the year. These rains also directly impact the phenology of many wild plants and agricultural operations, and are intrinsically tied to the function of quebradas, or seasonal creeks, that drain the region. As mentioned, severe and wide-reaching economic, environmental, and societal consequences can be realized in an abnormally dry rainy season. Thus, JFM is identified as the season of interest for this study."

**Comment:** p. 5. L. 2. *Referring to both EOF and PCA throughout. Stick with one to avoid confusion.*

**Response:** We apologize for the confusion and acknowledge that indeed EOF and PCA refer to the same process. Our intent in using both terms is to distinguish between the spatial patterns (EOF) and temporal trends (PCs) that come out of this process and provide complimentary information. Thus, we have opted to use both EOF and PC (and PCA as a descriptor of the process), and have also added a sentence to make this explicitly clear:

> "To evaluate the spatial and temporal patterns of regional precipitation, a principal component analysis (PCA) is performed on JFM seasonal precipitation totals (von Storch and Zwiers, 2001) based on data from the 29 stations. In PCA, a dataset is decomposed into orthogonal, uncorrelated modes representing distinctive signals, or variance, present in the dataset. PCA yields information describing both spatial patterns (empirical orthogonal functions, EOFs) and temporal trends (principal components, PCs) of variance experienced in the dataset…"

**Comment:** p. 5&6. Fig. 3&4. *It was not clear to me why PCA was used for the observed JFM precipitation totals? What purpose does it serve if the main target for the PCR model is for areal averaged precipitation totals anyway? Also, it is not stated what the physical interpretation of PC2 and PC3 are in this context (i.e. p. 9. L. 26. and p. 10. L. 9.). As it stands Fig. 3 does not really add anything. It is too difficult to see any difference the size of the red dots. Perhaps adding a scale and/or some gradual colour scale would help? Could you include what elevation threshold the topographic shading represents?*

**Response:** PCA was performed on JFM precipitation using all stations individually to understand how the first PC (explaining the most variance of all signals across the stations) compares with the station average across the region. The idea is to simply justify if the station average is a good representation of individual stations. In general, station averaged precipitation total correlates well with station-level data, however the variance experienced at each station is clearly not identical, as expected. We have clarified this in the text by adding:

> "… Additionally, the first principal component (PC) of the precipitation time series captures 51% of the variance in the data, and correlates well with station-averaged JFM seasonal precipitation observations (r = 0.99; Fig. 4). This exceptional level of correlation between the averaged observations and its first PC (as well as high levels of correlation between this first PC time series and individual station data) suggest that the station-averaged time series is an appropriate representation of regional precipitation."

Although the dominant signal present in this dataset correlates highly with the station-averaged time series, we do not mean to imply that higher modes of variance do not have an impact on regional precipitation. Referee #1 made a similar comment inquiring about the nature of PC2 and PC3. As per our response to Referee #1:

> The Referee highlights a good point, and we agree that it should be further clarified and discussed. The first mode clearly explains the majority of the variance in data (50%), and the second mode captures an additional ~20% of variance; however, the third drops to ~5%. Only these three modes are investigated in this study, for a cumulative total of 75% of variance captured. The manuscript has been revised to state the variance explained by each mode considered. Indeed, as suggested by the Referee, the second EOF suggests a dipole pattern; this description has also been added to the revised manuscript.

As referenced in the manuscript, Eklundh and Pilesjö (1990) postulated that high correlations between the first EOF of gridded precipitation and area averaged precipitation may suggest the presence of a large-scale climatic phenomena acting homogenously on regional precipitation. Additional studies that support this notion include Ogallo (1980), Mallants and Feyen (1990), Bisetegne et al. (1986). While we use a station average precipitation time series (and not area averaged, as noted by the Referee), the high correlation coefficient between this time series and the first PC of the original set of data may still be interpreted as a widespread homogenous influence on regional precipitation by a large-scale climatic phenomenon. With a correlation between the first principal component of regional JFM precipitation and JFM Niño 3.4 of -0.52, it is likely that this PC describes the modulation stemming from ENSO. As the Referee mentions, the subsequent PCs likely describe regional and local perturbations.

The studies mentioned in this response are now referenced in the manuscript. In addition to this response to Referee #1, the following passage has been added:

"While the first EOF likely illustrates ENSO's influence on regional precipitation, it is possible that higher order modes may describe other climatic and topographic forcings such as interconnected large-scale climatic phenomena or observed orographic effects. For example, the second EOF exhibits a dipole pattern, which may be related to the rain shadow phenomenon that causes the northeastern portion of the region to be wet and southwestern to be dry."

Finally, we removed Fig. 3 entirely and instead described the information more explicitly in text:

"… Even with significant changes in elevation across the region, the sign of the first EOF spatial pattern of all stations is negative (and at similar magnitudes) generally implying spatial homogeneity (Eklundh and Pilesjö, 1990) of JFM seasonal precipitation within this relatively small region. Additionally, the first principal component (PC) of the precipitation time series captures 51% of the variance in the data, and correlates well with station-averaged JFM seasonal precipitation observations (r = 0.99; Fig. 4)."

Figure numbers throughout the remainder of the manuscript have been revised accordingly.

**Comment:** p. 6. L. 12-13. *Am I correct in thinking none of the precipitation stations used here are within the rain shadow?*

**Response:** Technically, the Atacama Desert is the manifestation of the Andean rain shadow. While no precipitation stations used in this study are located there (although some are located at its edge), this portion of the Andes is unique in the sense that the mountain range is at its widest. Instead of an abrupt switch between wet and dry conditions as might be expected by some other notable rain shadows in the world, the Altiplano (and the majority of the stations used in this study) exists in a transitionary zone of sorts and exhibits a wet to dry gradient from northeast to southwest.

**Comment:** p. 7. Fig. 5: *Add units of SST anomalies (i.e. °C)? You could improve plot by adding two horizontal lines to represent El Niño/La Niña thresholds (i.e., ± 0.5°C). Also, define that you are using Pearson's correlation coefficient in the first instance (in the text) and define symbol as r. Then use r in every instance throughout the paper for clarity. I note this is done in some places and not in others (e.g.* p. 9. L. 30.*).*

**Response:** We appreciate the suggestion to include thresholds in Fig. 5; they have been added to the figure and are referred to in the text and figure caption as follows:

"… Strong El Niño (warm SST) conditions in the Niño 3.4 region are typically associated with drought in southern Peru, whereas La Niña (cool SST) conditions often align with wetter-than-average conditions (Fig. 5, El Niño and La Niña thresholds of 0.5°C and -0.5°C, respectively, included for context).

[Figure]

**Figure 5: Station-averaged JFM precipitation and concurrent JFM Niño 3.4 SST anomalies (r=-0.57). El Niño and La Niña thresholds marked with black solid line. During the period of record, 13 JFMs exceeded the El Niño threshold (0.5°C) and 18 years the La Niña threshold (-0.5°C)."**

We have also revised the manuscript to define our use of the Pearson correlation coefficient at its first instance and replaced subsequent references to correlation with the symbol $r$ where appropriate.

**Comment:** p. 10. Fig. 7: *Why use the first PC of regional JFM precipitation instead of just the area-averaged JFM precipitation total? I do note these are very similar and map would look the same.*

**Response:** We agree with the Referee that these two maps do look quite similar, and there may be motivation for simply presenting the station-average for simplicity; however, we have opted to present PC1 of the regional precipitation to highlight the dominant signal modulating precipitation. Again, this may look similar to the station-average, but it does not necessarily need to. Further, by evaluating PC1, we can identify distinct regions of SST, etc. that may lead us to better understand the controlling factors or phenomena (e.g. ENSO) that may be less obvious or evident using station-average. Then, this can be repeated for additional PCs to understand other (orthogonal) signals. Fig. 7 serves simply as an example of this type of analysis for predictor identification.

**Comment:** p. 12. L. 3. *"the JFM precipitation series"… but which one? First PC or observed totals?*

**Response:** The global wavelet analysis was performed on the JFM observed totals. We appreciate the clarification and have amended the manuscript to explicitly state this.

**Comment:** p. 13. Table 1. & L. 1-4. *The use of the asterisks is a little confusing. When I first looked at table 1 I presumed the asterisks was for statistically significant correlations. But are these instead those that are NOT correlated with JFM precipitation? Although you say that all are significant with at least one of the first three PCs. I can see here how perhaps the use of the first three PCs is useful but the reader is left with a bit of a jump to understand this without understanding what PC2 and PC3 represent. Could adding three additional columns to the right hand side of Table 1 for PC1, PC2, and PC3 help with this, then have the asterisks marking any value with statistically significant correlations. This allows the reader to see that perhaps one climate variable is correlated with all four precipitation series, or just e.g. PC3?*

**Response:** We apologize for the confusion and have revised Table 1 as well as the associated text.

"In total, 11 potential predictors are identified for prediction of station-averaged JFM precipitation based on previous literature and inference from spatial correlation maps, composite maps, and global wavelet analysis (Table 1). These potential predictors include both established climate indices and relevant regions of SST, SLP, and GH (as well as gradients of these variables).

All potential predictors included display a statistically significant correlation with at least one of the first three PCs of the station-averaged precipitation time series. In addition, five potential predictors are also statistically significant correlated with the station-averaged times series of precipitation, and marked with asterisks in Table 1.

**Table 1: The suite of potential predictors for JFM precipitation; correlations are based on JFM total precipitation and spatial averages across the regions noted, with statistically significant correlations marked with an asterisk.**

| Name | Large-scale climate variable | Timeframe | Region | | Corr. w/ JFM precip. | Most Correlated PC (Corr) |
|---|---|---|---|---|---|---|
| Niño 3.4 | SST | OND | 5° N-5° S | 170° W-120° W | -0.53* | PC1 (-0.52) |
| PDO | SST | OND | all areas north of 20° N | | -0.19 | PC2 (-0.35) |
| NP | SLP | D | 65° N-35° N | 160° E-140° W | -0.18 | PC3 (0.28) |
| WHWP | SST | OND | 28° N-8° N | 110° W-40° W | -0.16 | PC3 (-0.32) |
| | SST | OND | 0° -5° S | 160° W-140° W | -0.54* | PC1 (-0.54) |
| | SLP | D | 35° N-20° N | 150° W-135° W | 0.15 | PC2 (-0.36) |
| | SST gradient | OND | 0° -15° S (25° S-40° S) | 15° W-35° W (15° W-35° W) | 0.30* | PC2 (-0.29) PC3 (0.28) |
| | SST gradient | OND | 50° N-40° N (35° N-30° N) | 150° W-135° W (180° -165° W) | 0.38* | PC3 (-0.37) PC2 (0.27) |
| | GH 200 hPa | D | 10° S-15° S | 70° W-65° W | -0.35* | PC1 (-0.31) |

**Comment:** p. 14. L. 18-19. *Not clear how the ensemble in Fig. 10 was created. More detail needed here. Also, how many ensemble members etc.?*

**Response:** We agree that this may not have come across clearly in the original manuscript. Referee #1 made a similar comment in their review of the manuscript. As per our response to Referee #1:

Hindcasts for precipitation prediction are performed using principal component regression in a drop-one cross-validated mode. This includes – for each year of the hindcast – dropping the predictor data (Z) from the year being hindcasted, forming new PCs (and EOFs) conditioned on the remaining years, and fit to observations using multiple regression, providing an intercept coefficient, regression coefficients, and error term. The predictor data (Z) from the year dropped are then projected onto the EOFs to provide PCs for the dropped year. Finally, these PCs are multiplied by the appropriate regression coefficients and added to the intercept coefficient to provide a deterministic precipitation prediction for the dropped year. This is repeated for each year.

To create ensemble hindcasts, error terms from all years are assembled and a distribution is fit (using a kernel density estimator; the distribution is approximately Gaussian). For each hindcast year, 1,000 random draws from the distribution are added to the deterministic precipitation prediction to form an ensemble.

The manuscript has been revised accordingly to clarify this process.

**Comment:** p. 15. L. 6. *A few issues with this sentence. Suggest changing to something like: "An RPSS value less than zero signifies no forecast skill over the reference climatology forecast (i.e. it is ....), a value equal to zero for when the forecast is only as skillful as climatology, and values greater than zero represents a skillful forecast. A value of one represents a perfect forecast".*

**Response:** We appreciate the Referee's suggestion to restructure this sentence. The text of this passage now reads:

"An RPSS value less than zero signifies no forecast skill over the reference climatology forecast (i.e. the information provided by the developed forecast model is statistically less accurate than that provided by climatology), a value equal to zero for when the forecast is only as skillful as climatology, and values greater than zero represents a skillful forecast. A value of one represents a perfect forecast".

**Comment:** p. 15. L. 9-14. *Need more details about the use of 3x3 contingency tables, you might find the Svensson (2016) paper (and references therein) useful for this and as an example of statistical seasonal forecasting more generally. Also, more definition of what is meant by "extremely dry conditions". I know this is mentioned in the results, but it should be here that the methods details are given.*

**Response:** The authors thank the Referee for the reference, which has been added along with the following paragraph to clarify this point:

"… Results are presented in a three-by-three matrix, or contingency table, that illustrates the performance of the model for each category. Contingency tables are an alternative method of assessing the precision of model predictions that relies on categorical probabilities as opposed to simpler methods such as correlation (Svensson, 2016). Of particular interest in this study is the hit rate statistic, or the percentage of time the model accurately predicts (categorically) the actual observed condition. In addition, because prediction of regional drought is of particular interest, the likelihood of extremely dry conditions is also considered. For this case, extremely dry conditions are defined as station-averaged JFM precipitation totaling less than 250 mm, which occurs approximately 25% of the time, or approximately in 13 years across the time series."

**Comment:** p. 15. L. 16-18. Which combinations of the 11 predictors in Table 1 made it into the final PCR model? I know PCA was used, but can weight be given to original 11 predictor variables? For example, can I tell how important, if at all, Niño 3.4 is to the final model?

**Response:** The Referee raises a good point that we did not explicitly address in the manuscript. While we used the Generalized Cross-Validation (GCV) skill score considering all 11 potential predictors to determine the optimal number of PCs to retain as predictors (i.e., 4), we did not present results on how the inclusion or exclusion of subsets of the 11 variables changed overall skill performance. We have now partially addressed this in the revised manuscript by evaluating the difference in model skill for a second model construct, one in which Niño 3.4 and SST from 160° W-140° W 0° -5° S are dropped from the pool of potential predictors. While this is not exhaustive, these two predictors are the most highly correlated with JFM precipitation (and intrinsically are highly correlated with one another), and the differences in skill can give indication of the value in retaining this particular predictor – even though a PCA structure is used. The following text has been added to the Discussion section:

"The intent of this research was to explore the importance of including additional climate information in a predictive model that incorporates ENSO-based indices. To determine the importance of a model that does not include information regarding SST in the equatorial Pacific, a second hindcast model is developed using only 9 of the 11 original potential predictors. Niño 3.4 and SST from 160° W-140° W 0° -5° S are dropped in the modified model construct. Using the same cross-validated PCR methodology (as well as GCV to

determine the optimal number of potential predictor PCs to incorporate, i.e., 3), we produce hindcasts for the period of record (Fig. 13).

[Figure]

**Figure 13: Observed conditions for the period of record, as well as hindcasts produced using the original model (11 potential predictors, 4 PCs) and the modified model (9 potential predictors, 3 PCs).**

When comparing the results of the modified model to the original model, the importance of including ENSO in a model construct for precipitation prediction in southern Peru is highlighted. For example, the correlation coefficient between predicted conditions and observations for the modified model drops from $r = 0.58$ to $r = 0.37$ (still statistically significant, but skill reduced by roughly one-third). In addition, RPSS drops to only 0.05% from the original 16%, indicating that the information provided by the modified model is just barely more useful that that provided via climatology. In considering the hit-miss metric, the diminished skill of the modified model is also visible (Table 4).

**Table 4: Hit-miss matrix for the modified model with three equal categories: above normal (A), near normal (N), and below normal (B) precipitation.**

|  |  | Predicted conditions | | |
| --- | --- | --- | --- | --- |
|  |  | A | N | B |
|  | A | **6** | 10 | 1 |
| Observed conditions | N | 0 | **15** | 2 |
|  | B | 1 | 14 | **1** |

Above normal (A), near normal (N), below normal (B)

The modified model displays an evident bias towards predicting near normal conditions (more than 75% of the time). While the hit score of this model is reduced to 43%, more striking is the fact that the modified model produces an instance in which above normal conditions are prognosticated, but instead below normal conditions are experienced, arguably a more devastating forecast error in this region than the opposite situation (predicted below normal, experienced above normal). These metrics only reflect the critical importance of considering ENSO in regional precipitation prediction."

**Comment:** p. 16. L. 5-7. *The main message I get from Table 2 is that Near normal and Below normal precipitation is good, but it is the above normal that drags the hit rate to 51%.*

**Response:** This interpretation is accurate. Because of this, we explored the additional hit-miss matrix to evaluate above normal/near normal and extreme below normal precipitation predictions. Text has been added to the manuscript to explicitly state this interpretation.

**Comment:** p. 17. Sect. 7. *I like the idea of extended lead time analysis, but the technical details should be first outlined in the proposed 'Methods' section and results presented under a sub-heading within the 'Results' section.*

**Response:** As recommended by a previous comment, we have redistributed the technical details of the auxiliary applications to the 'Methods' section under appropriate subheadings. The results of these analyses are presented similarly in the 'Results' section.

**Comment:** p. 18. Fig. 12: *I'm missing how you are going from regional level to station level here. The extended lead time is good, but the spatial disaggregation is the weakest part of the analysis at present.*

**Response:** In an effort to more explicitly explain the spatial disaggregation portion of the study, the following text has been added to the manuscript:

> "… Using the regional-level categorical prediction probabilities for each year (above normal, near normal, and below normal; Fig. 12), ensemble predictions for each station are generated based on that station's own climatology. For example, the categorical probabilities at the regional-level for 2016 are predicted as 2% above normal, 7% near normal, and 91% below normal. For each station, JFM precipitation observations from all other years (excluding 2016) are randomly selected 1,000 times from that station's JFM precipitation distribution conditioned on the regional probabilities. Thus, the ensemble of predictions for that station for 2016 will have approximately 91% of its members from the below normal category, 7% near normal, and 2% above normal.

> The purpose of spatially disaggregating in this fashion is to maintain the statistical integrity of the regional-level prediction while reflecting appropriate magnitudes of precipitation experienced at different locations (Maraun, 2013). Station-level predictions are evaluated using the same metrics previously described."

**Comment:** p. 19. L. 6-9. *What statistical test is used to determine if the difference between regional and station correlation values is statistically significant or not?*

**Response:** Because the regional and station values are related, a dependent t-test for paired samples was used. The purpose of this statistical test is to search for significant changes or differences between two dependent variables. The goal of disaggregation was to produce a scaled precipitation forecast for local stations that maintained the statistical integrity of the regional prediction. Thus, a dependent t-test result of no statistically significant change is desired. We have clarified this in the revised manuscript.

**Comment:** p. 19. Sect. 8. *There is no discussion of the key limitations of the forecasting method/model (e.g. poor for above average precipitation). It would be good here to offer some suggested avenues for further research to overcome such methodological limitations.*

**Response:** We agree with this Referee comment. We will include a paragraph in the discussion section of the revised manuscript that explicitly discusses limitations, including poor performance in predicting above average precipitation conditions, the regional versus local nature of predictions and associated skill, real-time data requirements and trade-offs with longer prediction lead time, etc. In addition, we will outline potential avenues for further research such as alternative modeling approaches and integration with hydrology and other sectoral/decision-making models.

**Comment:** *I like the final paragraph on* p. 20 *as it highlights well the practical importance of seasonal forecasting using climate information in a region where none is currently available.*

**Response:** We truly do hope that the work undertaken by our group will be of practical use to stakeholders in the region. To clarify, though, it was not our intention to frame this study as the Prometheus of seasonal forecasting for the people of southern Peru. Certain national and local entities (both public and private) currently do employ forecasting techniques to predict regional precipitation to varying extents. For example, SENAMHI uses Niño index-based forecasts and SPCC has explored and developed predictive models that employ artificial neural networks. Instead, the purpose of this study is to expand on the region's existing capacity to predict precipitation. The manuscript has been edited to reflect this sentiment in a more appropriate and explicit manner.

**Minor Comments (all accepted and corrected in the manuscript)**

p. 1. L. 28. Change "vary drastically" to e.g. "vary considerably"

p. 2. L. 19. Change "wreaked havoc" to e.g. "was particularly severe"

p. 2. L. 26. Change "The dire" to e.g. "The societally challenging"

p. 4. Fig. 2. Add "…using data from 29 precipitation stations in Sect. 2" to caption.

p. 5. L. 7. The Eklundh and Pilesjö (1990) reference is for Ethiopia. Should reference go after "homogeneity" instead?

p. 6. L. 11. Add a comma in (Garreaud, 1999)

p. 8. L. 5-8. Very short paragraph, better added to the previous one

p. 9. L. 3. Change "hydrometeorologic" to "hydrometeorological"?

p. 9. L. 21. Change "previously identified" with "identified in Sect. 3"

p. 10&11. Fig. 7&8. Worth adding a dot/circle to mark study region on the maps for an international audience?

p. 11. L. 1-5. Delete section as repeated from p. 10.

p. 13. Table 1. Add space between 'Time frame'

p. 18. L. 9. Add full stop after "…etc.)"

p. 1. L. 19. Add "regional" and "totals" in front of and after "January-March precipitation", respectively.

**References**

Maraun, D.: Bias Correction, Quantile Mapping, and Downscaling: Revisiting the Inflation Issue, J. Climate, 26, 2137–2143, 2013.

Svensson, C.: Seasonal river flow forecasts for the United Kingdom using persistence and historical analogues, Hydrol. Sci. J., 61(1), 19–35, 2016.

Van Loon, A. F.: Hydrological drought explained, Wiley Interdiscip. Rev. Water, 2(4), 359–392, 2015.

Xu, C.: Climate Change and Hydrologic Models: A Review of Existing Gaps and Recent Research Developments. Water Res. M., 13, 369-382. 1999.